palaeontology

Ediacaran, *Arumberia*, streamers, matground, non-marine, terrestrial

**Author for correspondence:**
S. McMahon
e-mail: sean.mcmahon@ed.ac.uk

# Late Ediacaran life on land: desiccated microbial mats and large biofilm streamers

S. McMahon[1,2], J. J. Matthews[3], A. Brasier[4] and J. Still[4]

[1]UK Centre for Astrobiology, School of Physics and Astronomy, University of Edinburgh, James Clerk Maxwell Building, Edinburgh EH9 3FD, UK
[2]School of Geosciences, Grant Institute, University of Edinburgh, Edinburgh EH9 3FE, UK
[3]Oxford University Museum of Natural History, Parks Road, Oxford OX1 3PW, UK
[4]School of Geosciences, University of Aberdeen, King's College, Aberdeen AB24 3UE, UK

SM, 0000-0001-8589-2041; JJM, 0000-0001-5350-8439; JS, 0000-0003-0249-3876

The Ediacaran period witnessed transformational change across the Earth–life system, but life on land during this interval is poorly understood. Non-marine/transitional Ediacaran sediments preserve a variety of probable microbially induced sedimentary structures and fossil matgrounds, and the ecology, biogeochemistry and sedimentological impacts of the organisms responsible are now ripe for investigation. Here, we report well-preserved fossils from emergent siliciclastic depositional environments in the Ediacaran of Newfoundland, Canada. These include exquisite, mouldically preserved microbial mats with desiccation cracks and flip-overs, abundant *Arumberia*-type fossils and, most notably, assemblages of centimetre-to-metre-scale, subparallel, branching, overlapping, gently curving ribbon-like features preserved by aluminosilicate and phosphate minerals, with associated filamentous microfossils. We present morphological, petrographic and taphonomic evidence that the ribbons are best interpreted as fossilized current-induced biofilm streamers, the earliest record of an important mode of life (macroscopic streamer formation) for terrestrial microbial ecosystems today. Their presence shows that late Ediacaran terrestrial environments could produce substantial biomass, and supports recent interpretations of *Arumberia* as a current-influenced microbial mat fossil, which we here suggest existed on a 'streamer–arumberiamorph spectrum'. Finally, the absence of classic Ediacaran macrobiota from these rocks despite evidently favourable conditions for soft tissue preservation upholds the consensus that those organisms were exclusively marine.

## 1. Introduction

As participants in primary production, biological and oxidative weathering, nutrient cycling, river channel stabilization, clay mineralization and pedogenesis, late Precambrian terrestrial (tidal, fluvial, lacustrine and soil) ecosystems may have been important agents of global biogeochemical change (e.g. [1–4]). However, their fossil and geochemical records are sparse and somewhat controversial (e.g. [5]). While claims have been made for a terrestrial Ediacaran macrobiota (e.g. [6]), this conflicts with sedimentary evidence indicating that all known Ediacaran macroorganisms resided within marine environments [7–12]. Putative microbially induced sedimentary structures (MISS) are, however, commonly reported in late Ediacaran tidal and fluvio-tidal siliciclastic sediments worldwide (see tab. 1 in [13]).

Here, we report fossil terrestrial life from the late Neoproterozoic siliciclastic rocks of the Signal Hill Group, Avalon Peninsula, Newfoundland, Canada. The Avalon Peninsula preserves a more than 7.5 km thick late Neoproterozoic

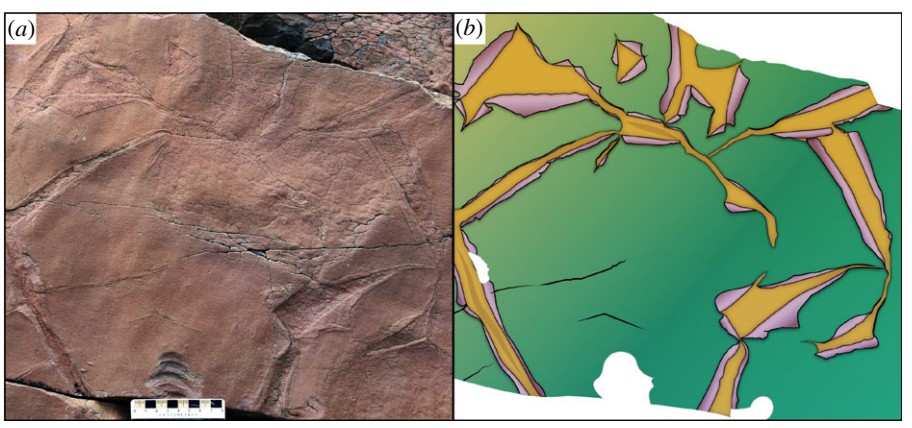

**Figure 1.** Exquisite mouldic preservation of a microbial mat with desiccation features. (*a*) Photograph of a mudstone veneer capping a sandstone block in the Gibbett Hill Formation at Bear Cove Point, Newfoundland. Scalebar is in centimetres. (*b*) Reconstruction of the microbial mat showing sharply defined overfolded crack margins and exposed under-mat layer. (Online version in colour.)

volcano-sedimentary succession deposited close to the shoreline of the micro-continent Avalonia [14] (electronic supplementary material, figure S1). The succession shows an overall shallowing-upwards trend from deep-marine basin-floor facies to shallow marine and ultimately emergent siliciclastic units [8,15,16]. The uppermost Signal Hill Group records the transition to non-marine conditions. Coastal exposures in the area around Ferryland span a stratigraphic thickness of approximately 1.5 km (electronic supplementary material, figure S1). In this area, the basal Cappahayden Formation and overlying Gibbett Hill Formation represent deposition within shallow marine, wave-influenced and increasingly sand-dominated environments [17]. The up-section gradational transition from the thickly bedded grey cross-stratified and rippled sandstones of the Gibbett Hill Formation into the Ferryland Head Formation, defined by the appearance of red sandstones [15], records the onset of periodically emergent deposition [17]. The 'E' Surface at Mistaken Point (electronic supplementary material, figure S1), famous for its rangeomorph fossils (a clade of soft-bodied organisms comprising one or more fractal-like 'fronds'), has recently been dated to 565.00 ± 0.64 Ma [18]. While the units cropping out in Ferryland are more than 1.5 km stratigraphically above the 'E' Surface, they are also 50 km to the north and were deposited in a southwards-prograding system [19]—note that this diachroneity could imply an age closer to approximately 565 Ma than might otherwise be expected. In any case, the Signal Hill Group pre-dates the Ediacaran–Cambrian transition (e.g. [8,20–22]). While the upper boundary of the Group does not crop out in the area, a correlative section in the Musgravetown Group is unconformably overlain by the Lower Cambrian Random Formation [19]. Here, we report the presence of exquisite mouldically preserved desiccated microbial mat fossils in the Gibbett Hill Formation, abundant *Arumberia*-type fossils in the Ferryland Head Formation, and associated metre-scale structures we interpret as fossil biofilm streamers induced by currents.

## 2. Results and interpretation

### (a) Desiccated microbial mats with 'flip-overs'

The Gibbett Hill Formation has previously been described as a succession of thickly bedded grey and buff sandstones deposited in shallow marine environments [15,17]. At Bear Cove

Point (46°56′26″ N, 52°53′31″ W), within approximately 40 m of the base of the Formation at a site and stratigraphic position where fossils have not previously been discovered, we observe siltstones and mudstones interbedded among the more abundant sandstones. These rocks host sharp-crested, symmetrical ripple marks, which we interpret to record wave activity, as well as multiple horizons with sedimentary cracks suggestive of desiccation (electronic supplementary material, figure S2). At this locality, we report fossil microbial matgrounds, exceptionally well preserved in epirelief on sandstone bedding planes capped by mudstone veneers. One such surface displays a semi-polygonal network of shallow fossil shrinkage cracks, which are bordered by slightly raised, flat-topped strips and wedges (electronic supplementary material, figure S3; figure 1*a*). Most of the cracks are at the decimetre scale in length and open to a width of several centimetres; they are preserved with sub-millimetric epirelief. The cracks are bordered by slightly raised, flat-topped strips and wedges that have sharply defined but irregular edges and are consistently just under half the diameter of the cracks they border. Based on comparisons with modern tidal microbial mats (electronic supplementary material, note S1) and with fossil MISS (e.g. [23]), we interpret these raised features as the results of the desiccation-shrinkage of smooth, ductile, coherent, epibenthic microbial mats exposed to air. Shrinkage caused the crack margins to peel back, widening the cracks to several centimetres and forming inverted flaps ('flip-overs' *sensu* [24]) that are retained in sub-millimetre detail on the bedding plane (figure 1*b*). Thus, the relief of the bedding plane preserves a mould of the top of the mat including these flaps, rather than simply the impression made by the mat on the underlying sediment, as in some MISS. To our knowledge, such well-preserved examples of microbial mat flip-overs have not previously been found in the siliciclastic fossil record of any period, although less well-preserved examples are reported (e.g. from the Palaeoproterozoic [24,25], Mesoproterozoic [26] and Cambrian [27]). A variety of related structures such as 'jelly rolls', 'roll-ups' and desiccated mats without clear flip-overs have also been described from the Precambrian fossil record (e.g. [25,28–30]). At Bear Cove Point, the open cracks also reveal a delicately reticulate surface texture that we identify as the 'elephant skin texture' MISS, well known from under-mat (palaeo)surfaces [24,31,32].

Noting previous shallow marine interpretations of the Gibbett Hill Formation, the presence of wave ripple marks

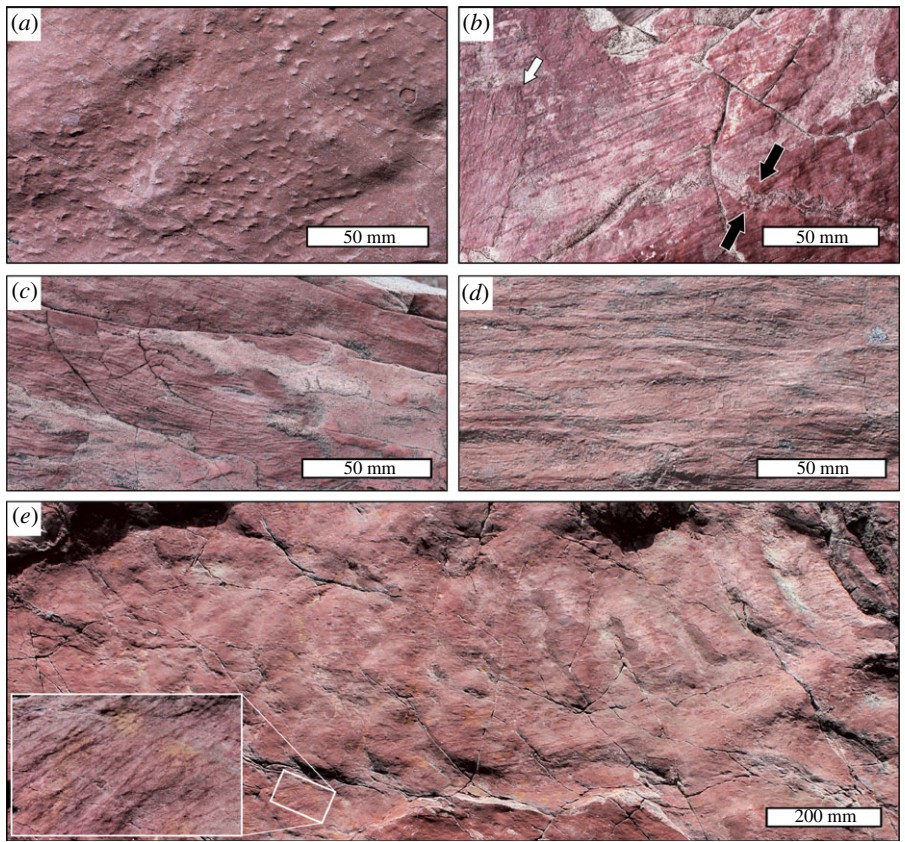

**Figure 2.** Sedimentary structures on bedding planes in the Ferryland Head Formation, siltstone facies. (*a*) Angular pseudomorphs after gypsum in a mudstone lamina. (*b*) *Arumberia*, a putative microbial matground fossil comprising subparallel curving ridges (rugae), which are here oriented lower-left to upper-right. Black arrows indicate a sand-filled desiccation crack in the otherwise argillaceous lamina. The white arrow indicates the edge of a second, overlying *Arumberia* layer. (*c*) *Arumberia*morph texture with linear ridges approximately 0.5 mm in diameter, some of which overlap each other. (*d*) A more coarsely lineated, somewhat ribbon-like arumberiamorph texture. (*e*) A sandstone surface showing gentle undulations best interpreted as ripple marks (horizon S2, electronic supplementary material, figure S5). A highly branching form of *Arumberia* is preserved as an argillaceous red veneer, with lineations consistently aligned perpendicular to the ripple crests (i.e. in the same direction as the inferred palaeocurrent). Inset: Close-up view of area indicated, showing complex nature of the lineated, arumberiamorph surface texture. (Online version in colour.)

and desiccation cracks on multiple horizons, and the new evidence for emergence provided by these dessicated microbial mats, it seems likely that the latter inhabited a tidal setting. The Signal Hill Group as a whole has previously been described as a 'shallow marine to proximal alluvial fan' succession [21]. Our results refine this model, revealing at least one cycle of deepening and then shallowing after the deposition of the emergent features described here.

## (b) *Arumberia* and associated structures in the Ferryland Head Formation

Fossil surfaces were studied at Ferryland Head on the eastern coastline of the Avalon Peninsula, Newfoundland, Canada. A roughly 10 m thick succession was studied on the northern side of a cove (47°01′13″ N, 52°51′26″ W), situated directly to the north of the lighthouse at Ferryland Head, and south of Bread and Cheese Cove (see electronic supplementary material, figure S4 for a photographic overview, and electronic supplementary material, figure S5 for a log). This unnamed cove provides both bedding plane and bed-section outcrops of the Ferryland Head Formation, with strata being easily traced along strike for distances of more than 100 m. At this locality, the stratigraphy can be broadly divided as follows. (i) A buff-coloured sandstone facies comprising thickly bedded arkosic sandstones, with occasional tabular cross- and convolute-

lamination. Beds are laterally continuous, very gradually pinching out to leave sandstone bodies that can be greater than 100 m wide. (ii) A red-brown, comparatively siltstone-rich facies comprising thinly interbedded red-brown mudstones, siltstones and buff sandstones, in some places showing normal grading. Sandstone laminae in this facies are generally no thicker than 3 cm, with finer-grained red-brown coloured laminae measuring 1–3 mm thick. Convolute-lamination is not uncommon, with desiccation cracks, and current and wave ripples also being found. The studied roughly 10 m thick succession is dominated by this red-brown siltstone-rich facies.

Previous summaries of the sedimentological evidence [15,33] suggest that the Ferryland Head Formation was deposited in non-marine conditions. We consider the laterally extensive bed geometries and observed sedimentary structures to be indicative of deposition within a braided channel system. The buff sandstone facies is here considered to be associated with the channel and proximal overbank environments, and is observed to transition upwards into the finer-grained facies. These red-brown interlaminated units are interpreted to represent braidplain environments of unconfined deposition, influenced by periodic wetting and drying cycles (producing sedimentary pseudomorphs after gypsum, and desiccation cracks; figure 2*a*,*b*). Wave ripples are found within this facies, potentially representing deposition in ephemeral braidplain lakes subject to agitation by

wind [34]. Current ripples are likely to be associated with decelerating unconfined flows [35]. Further work is required to assess the extent of tidal influence on these braidplain environments, and therefore the salinity of the palaeohabitat represented by the fossils reported here.

The studied approximately 10 m thick succession is dominated by the red-brown siltstone-rich facies, which on reddish mudstone horizons preserves diverse microbial mat textures, in some instances cross-cut by sand-filled desiccation cracks (e.g. figure 2*b*). Many of these textures can be identified with known varieties of *Arumberia*, a problematic sedimentary fabric associated with Ediacaran matgrounds and characterized by a distinctive, lineated surface texture (see electronic supplementary material, note S2 for discussion of *Arumberia* nomenclature). *Arumberia* is well known from late Ediacaran and early Cambrian intertidal, deltaic and alluvial plain siliciclastic facies worldwide, including desiccated sediments (e.g. [36–41]). The classic form designated *Arumberia banksi* consists of sets of curving, subparallel, equidistant, bifurcating ridges (rugae) several centimetres in length, which fan out over bed surfaces. At Ferryland Head, arumberiamorph textures vary in scale (e.g. figure 2*c,d*) and range from *Arumberia banksi* with well-defined parallel 'rugae' (albeit with minimal 'fanning out'; e.g. figure 2*b*) to more irregular, branching ridges (e.g. figure 2*e*; cf. *Arumberia vindhyanensis* [39]). These features are generally oriented parallel to the palaeocurrent direction inferred from current ripples in the sandstone immediately underlying them (e.g. figure 2*e*). The same phenomenon has been noted in respect of *Arumberia* rugae at other localities globally [36,37,40–42].

## (c) Argillaceous ribbons interpreted as fossil biofilm streamers

A number of bedding planes at Ferryland Head display multiple, elongate, subparallel, gently curving argillaceous epirelief structures (ribbons) approximately 1–20 mm in diameter. Some examples are directly associated with (and parallel to) *Arumberia* rugae preserved in the same fashion. The ribbons that are largest and least similar to previously reported morphologies of *Arumberia* are found on an individual bedding plane exposure of approximately 4 m$^2$, characterized by a smooth, uneven surface, probably shaped by currents. This bedding plane, 'S1', is found within a section of the siltstone-rich facies on the northern side of the cove (electronic supplementary material, figure S4). The numerous well-defined ribbons are composed of red-brown clay and silt, contrasting visually with the underlying buff-coloured fine sandstone; some individual ribbons can be traced for more than a metre before tapering out, becoming indistinct or terminating in erosional effacement (figure 3*a,b*). Except where weathered flat, many of the ribbons protrude slightly from the bedding plane in positive epirelief.

In plan view, ribbons show a range of enigmatic morphological features (figure 3*c–h*). They commonly fork into two or more thinner branches, which typically diverge at low angles (e.g. figure 3*c,d*). Rarer high-angle branching is also observed (figure 3*e*). Some ribbons are connected in complex anastomosing webs (e.g. figure 3*b,f*) and several appear to cross over each other (within the same lamina; e.g. figure 3*g*). One ribbon contains a discrete, very pale pink, longitudinally striated portion approximately 30 mm long, which protrudes convexly upwards from the bedding plane and evidently

differs in material composition from the remainder of the ribbon, which is argillaceous (figure 3*h*). To diagnose the composition without damaging this significant fossil surface, we sampled and thin-sectioned loose material of identical colour and lustre found within 1 m stratigraphically. Energy-dispersive X-ray (EDX) spectroscopy showed that the light pink material was dominated by a phosphate mineral, probably fluorapatite (electronic supplementary material, figure S6).

We consider these megascopic ribbons to be probable fossils. Known abiotic processes are unlikely to have produced the observed structures, for the following reasons. The features are roughly perpendicular to a poorly defined set of ripples, which rules out the shrinkage crack feature 'Manchuriophycus', which has a sinuous shape running along ripple troughs. Microchannels and flute marks, while expected to be parallel to flow, are negative rather than positive epirelief features. Glacial striae are known from the region, but would be more consistently straight, parallel, groove-like features, not positive, ribbon-like features that branch and curve. Thin section evidence also rules out the above abiotic interpretations, and shows that these features are not the bedding expression of tectonic cleavage or fracture networks (figure 4*a,b*). Thin section and bed section views show that the ribbons are localized to particular horizons, and lack the laminae-intersecting infill sediments seen in sedimentary structures such as syneraesis cracks, desiccation cracks and dewatering structures. It might be suggested that the ribbons on S1 are the product of selective modern weathering but thin sections demonstrate that the structures continue beneath the exposed surface (e.g. figure 4*b*); moreover, similar structures are not observed on the multitude of comparably aligned surfaces. Weathering may have enhanced the visual contrast between the better indurated, clay-rich material composing the ribbons and the surrounding lithology. The three-dimensional preservation of the ribbons in argillaceous material (e.g. figure 4*a,b*), their large size, and their wide spacing rule out interpretation of the structures as being caused by parting lineation, which is only found in sandstones.

Micropalaeontological observations and taphonomic considerations (electronic supplementary material, note S3) also support our interpretation of the ribbons as biogenic. The phosphatic replacement seen in one section of a ribbon, and the occurrence of stratigraphically proximal phosphatic concretions, imply the liberation of phosphate ions from decaying organic matter (e.g. [43]). In addition, although the S1 surface could not be sampled, similar material sampled 5 cm higher in the stratigraphy was thin-sectioned at a low angle to bedding, revealing that clay- and phosphate-rich laminae contained filamentous structures that we cautiously interpret as probable fossil microorganisms (figure 4*c*). These filaments, like the ribbons themselves, appear to have been preserved three-dimensionally by authigenic clay replacement (aluminosilicification), in common with *Arumberia* at Ferryland Head and at other localities globally (e.g. [40]). Indeed, early replacement of soft tissues by clay minerals has been implicated in the preservation of several important Proterozoic and Phanerozoic Lagerstätten (e.g. [44,45]). Our observations do not necessarily constrain the timing of clay replacement at Ferryland Head (see electronic supplementary material, note S3 for discussion). Nevertheless, the argillaceous composition of the ribbons and arumberiamorph structures is clearly consistent with an organic origin.

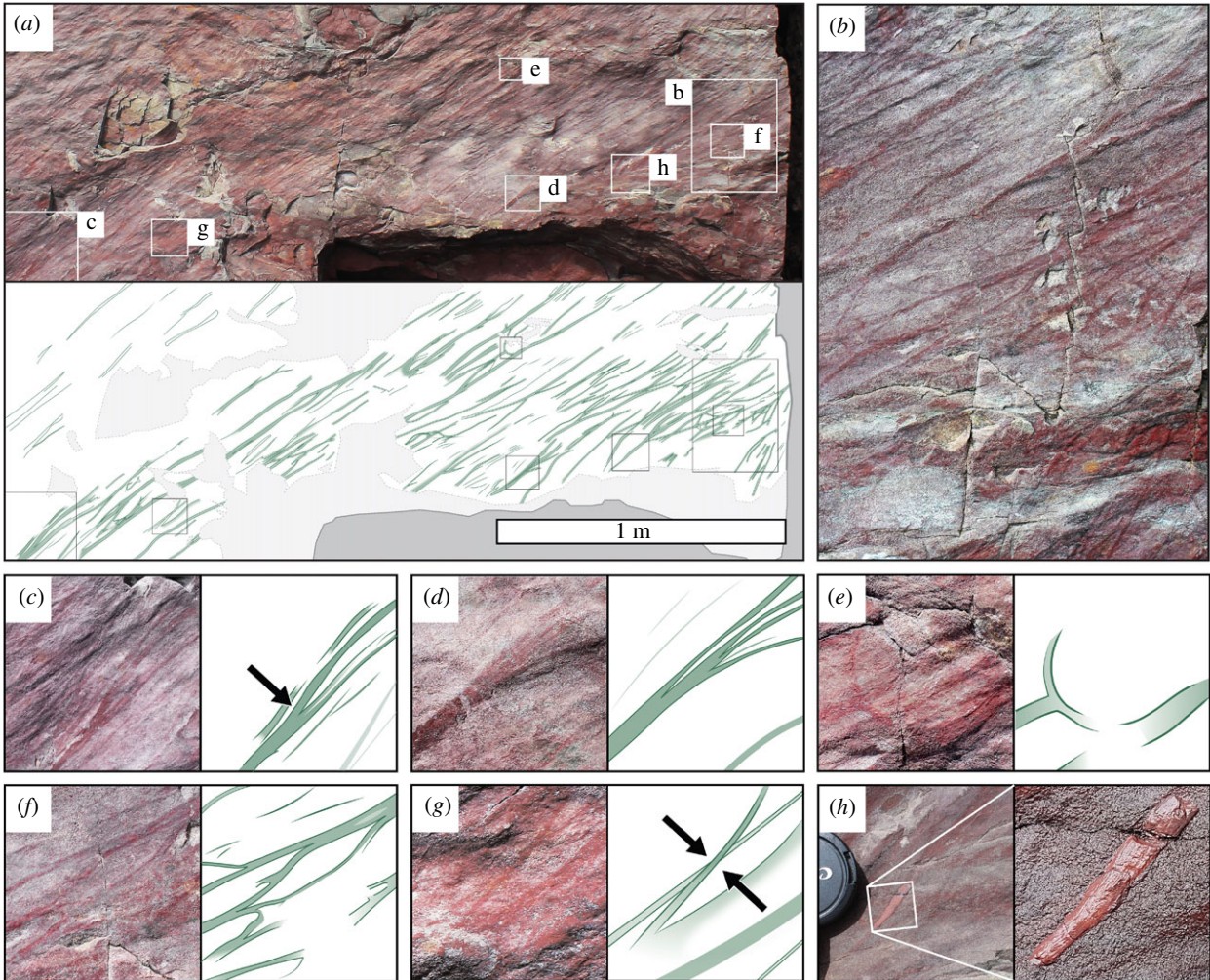

**Figure 3.** Argillaceous ribbon-like structures on a sandstone bedding plane at Ferryland Head (horizon S1, electronic supplementary material, figure S3). (*a*) Field photograph and interpretative sketch (based on multiple photographs) of a fine sandstone bedding plane assemblage of reddish argillaceous ribbon-like structures showing the positions illustrated by figure (*b*–*h*). (*b*) Photograph of the same surface with a narrower field of view, showing multiple branching and converging ribbons. (*c*) Photograph and interpretative sketch of a low-angle bifurcation (arrowed). (*d*) Photograph and interpretative sketch of a low-angle multifurcation. (*e*) Photograph and interpretative sketch of a curved, high-angle bifurcation. (*f*) Photograph and interpretative sketch of interconnected ribbons. (*g*) Photograph and interpretative sketch showing one ribbon crossing over another at arrowed position. (*h*) Ribbon with light coloured, 30 mm long, convex portion; close-up shows longitudinal lineations visible when wetted. Photographs have been enhanced for contrast.

The irregular, anastomosing, web-like morphology of the ribbons (e.g. figure 3*b*) indicates that they are not large ribbon-shaped individual macroalgae (e.g. vendotaeniaceans). Rather, we interpret them as compressed, aluminosilicified, partly phosphatized, microbial/algal 'streamers'. Streamers are string-, ribbon- and rope-like biofilm structures composed of microbes (and/or macroscopic algae) and their extracellular organic substances, ranging from tens of micrometres to several metres in length. They are commonly found today in very shallow, gently flowing waters, especially those influenced by eutrophication, acid mine drainage or thermal springs (e.g. [46–48]). Constituent organisms commonly include cyanobacteria, sulfur-oxidizing bacteria, thermophilic bacteria (e.g. phylum Aquificales) and green algae [48–51]. Streamers are tethered to the sediment–water (or rock–water) interface at one end, commonly in a benthic microbial mat, and extend downstream, bifurcating and splaying out in the same direction as the current. Thus, streamers differ from mat wrinkles or ridges in being partly detached structures suspended in the water column or resting lightly on the sediment surface. Large streamers tend to be widely spaced but can converge into complex anastomosing networks (figure 5*a*,*b*), while smaller streamers commonly form turf-like outgrowths of filamentous mats (figure 5*c*,*d*).

## (d) The streamer–arumberiamorph spectrum

The co-occurrence at Ferryland Head of megascopic fossil streamers and arumberiamorph mat textures, both preserved mostly by clay replacements, informs our interpretation of both phenomena. Reconstructions of *Arumberia* as a discrete macroorganism (e.g. [52]) are implausible given its indistinct margins and the gradational transitions observed between different morphologies over short distances [41]. Rather, we agree with previous authors that *Arumberia* represents an organic matground morphotype arising from the interaction of dense biotic communities, siliciclastic sediment and flowing water [13,37,39,41,42]. Partial modern analogues may include regularly spaced wrinkles and ridges observed on cyanobacterial mats in ponds [42], current-parallel microbial wrinkles on rippled tidal flats [53] and sub-millimetre, current-parallel ridges produced in laboratory experiments using fluvial biofilms [54]. However, we propose that the raised ribbons and lineations on fossil arumberiamorph

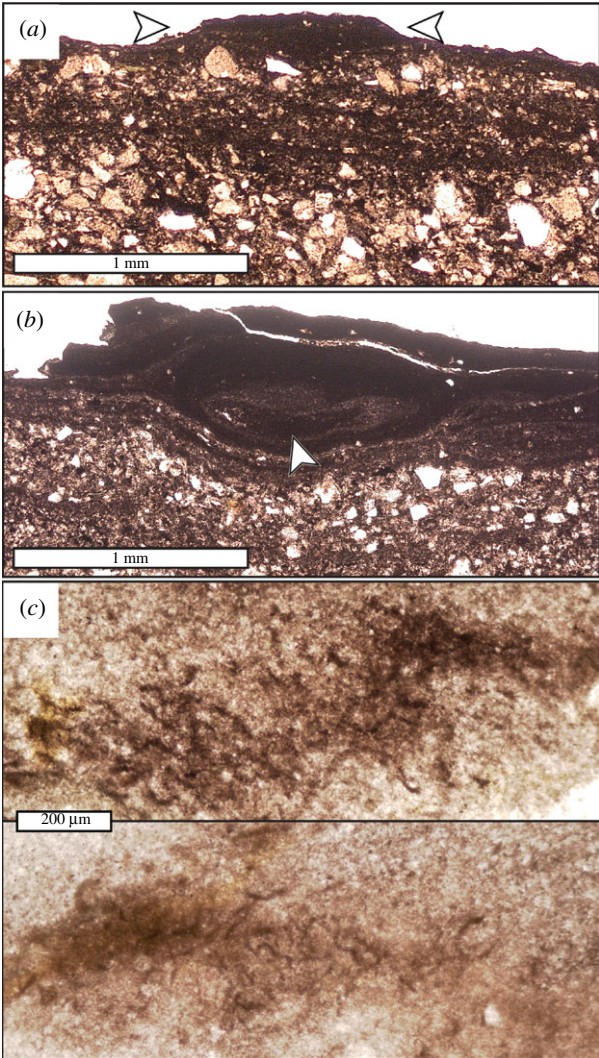

**Figure 4.** Photomicrographs of small argillaceous ribbons (*a,b*) and poorly defined filaments (*c*) in thin sections from Ferryland Head. (*a*) Transverse section through small raised argillaceous ribbon (arrowed) interpreted as a fossil microbial streamer from horizon S2, occurring on bedding plane (electronic supplementary material, figure S5; figure 2*e*). Thin section FRY-10. (*b*) Transverse section through a second streamer (arrowed) on the same bedding plane as A (thin section FRY-10), mantled with silt and sandwiched between argillaceous laminations to form a concentric bull's-eye pattern. Argillaceous laminae above and below the streamer were deflected by silt and/or authigenic cements during compression. (*c*) Two views of thin section NL-3 cut at low angle to bedding, showing poorly defined filaments that we interpret as probable microfossils. Note preferred orientation of filaments in the upper image. Sample collected approx. 5 cm above S1 streamer-bearing horizon. (Online version in colour.)

surfaces include not only ridges (or 'rugae') but also the remains of streamers (which may be identical with the 'cords' described by McMahon *et al.* [43], who report positive and negative epirelief examples from NW France). These streamers may usually have been smaller and more mat-like than the megascopic ribbons dominating horizon S1 at Ferryland Head (e.g. compare figures 2*c* and 5*d*). It can be difficult to discriminate between streamers and linear mat thickenings from preserved epirelief alone, but thin sections orthogonal to the lineation direction can provide decisive evidence. Ridges (rugae) manifest as laminated clay-rich, tent-like convexities (e.g. see fig. 4C in [42]). By contrast, streamers form discrete, rather flat, argillaceous ribbons (figure 4*a*; sample is from horizon S2, also shown in figure 2*e*). Ribbons can also exhibit

a compressed bull's-eye pattern with a clay-rich core and a mantle of silt, draped by overlying clay laminae (figure 4*b*, from the same surface). This internal grain-size patterning appears to be topologically incompatible with a 'rugose' thickening or wrinkle in a microbial mat, but is explained by our interpretation of these features as streamers that lived in partial suspension and were buried with a passively acquired coating of silt. Biomass, both streamer and microbial mat, was then apparently replaced by clay.

The associations observed at Ferryland Head suggest that *Arumberia* matground structures and streamers of all sizes do not represent different organisms but, rather, a spectrum of patterns of spatial organization manifested by substantially similar communities. The affinities of these organisms are not obvious; comparison with modern arumberiamorph mats [42] and large modern streamers suggests that they were autotrophic (chemotrophic or phototrophic) and could have been largely (cyano)bacterial, but we cannot exclude the possibility that eukaryotes were also present. The filamentous microfossils associated with the streamers (figure 4*c*) are consistent in size and morphology with both cyanobacteria and eukaryotic microalgae. The mode of preservation by clay minerals may result from bacterially mediated reactions but is consistent with either bacterial or eukaryotic affinity for the fossils themselves [55–57]. The discovery of ribbon-like microbial communities preserved in a terrestrial lithology has at least one precedent in the form of millimetre-to-centimetre-scale, carbonaceous 'strap-shaped' cyanobacterial fossils from the early Silurian of Virginia [58], suggested here to be possible fluvial streamers.

The diversity of macroscopic arumberiamorph textures and streamers at Ferryland Head was probably shaped by variations in hydrodynamic conditions such as shear stress and turbulence. These responses may have fallen on a continuum, with weak currents supporting the development of 'rugose' arumberiamorph textures, while stronger currents drew out suspended streamers (figure 6) given enough time. This 'streamer–arumberiamorph spectrum' model sheds new light on occurrences of *Arumberia* elsewhere in the world. For example, many specimens of *Arumberia banksi* show fan-shaped patterns and small, scattered domes; both features that can be observed in modern streamer mats (e.g. figure 5*d*). The millimetre to centimetre-wide longitudinally striated ribbons of *Arumberia ollii*, and the parallel or fanning, 0.5 mm-diameter curved filaments of *Arumberia beckeri*, both reported as aluminosilicified compressions from the Ediacaran of the Central and South Urals by Kolesnikov *et al.* [40], can now also be interpreted as small streamers similar to those at Ferryland Head. Kolesnikov *et al.* [40] suggested that these two forms might be taphonomic variants of each other, and Kolesnikov *et al.* [42] proposed that they were both unrelated to *Arumberia* since they consist of discrete elements rather than textures on a mat. Our model suggests that they probably belong on a spectrum with arumberiamorphs, and predicts that intermediates may exist. We also note that streamer-mats preserved on bedding planes have previously been reported from Devonian hot spring sinter deposits; these define a surface texture aligned to and preserved within palaeo-channels that may be similar to the textures observed at Ferryland Head, and show a finely laminated structure in cross-section [59]. Our results, and our reinterpretation of *Arumberia*, extend the fossil record of terrestrial streamer formation into the Precambrian.

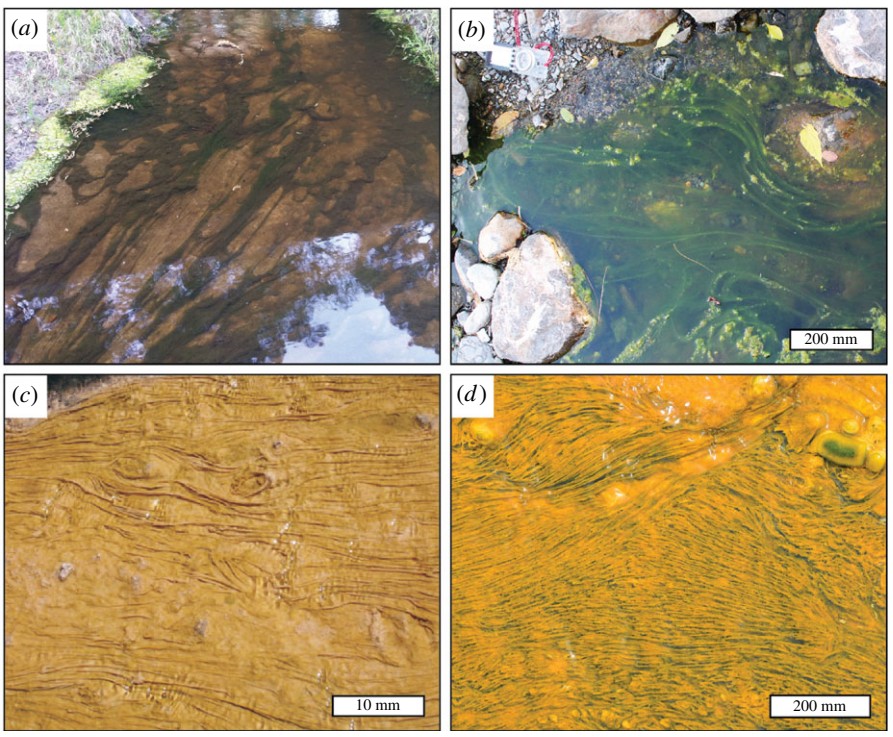

**Figure 5.** Modern examples of shallow-water algal and microbial streamers. (*a*) Long, ropy green streamers in a shallow stream (1–2 m across) near Lufkin, TX, USA. Photograph courtesy of Angelina and Neches River Authority Clean Rivers Programme. (*b*) Sinuous green streamers in a rocky, gently flowing stream near Deiva Marina, Liguria, Italy. Individual streamers diverge, reconverge and anastomose. (*c*) Small streamers in fast-flowing hot springs at the El Tatio geyser field, Chile. Photograph courtesy of Gary Smith, Geothermal Scientific Investigations Ltd. (*d*) Streamer-mat in fast-flowing hot springs at Orakei Korako, New Zealand. Photograph courtesy of Gary Smith, Geothermal Scientific Investigations Ltd. The fanning-out behaviour and pustule-like associated structures are comparable to many instances of *Arumberia*. (Online version in colour.)

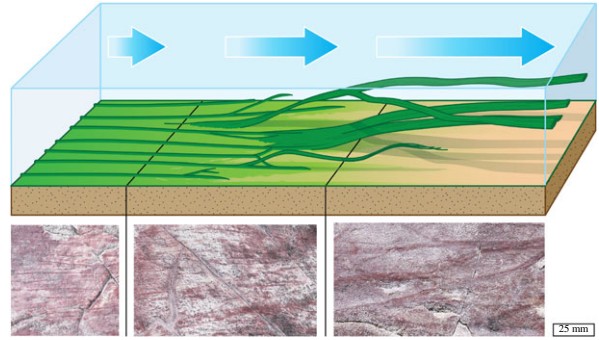

**Figure 6.** Conceptual model of Ferryland Head 'streamers' and relationship to *Arumberia*. The cartoon shows how increasing shear stress or other hydrodynamic forcing (from left to right) may explain the emergence of ribbon-like streamers from 'rugose' arumberiamorph microbial matgrounds. This model accounts for the occurrence of intermediate states characterized by a superposition of both morphotypes seen at Ferryland Head (photographs below, with scale bar). (Online version in colour.)

## 3. Discussion

The exceptionally preserved flipped-over desiccated microbial mats preserved at Bear Cove Point, as well as the ribbon-like current-induced streamers at Ferryland Head, provide new windows onto late Ediacaran life on land. Photosynthetic algal and microbial biofilms and streamers commonly dominate primary production in streams and rivers today; their biomass is held in check largely by grazing animals [51]. Streamers can also be found in tidal settings (e.g. fig. 9C in [60]) but are often most abundant where animal activity is suppressed, as in hot springs and acidic streams [61,62].

Although a profusion of microbial and algal life can be associated with high nutrient fluxes, the abundance of arumberiamorph mats and streamers at Ferryland Head—and of arumberiamorphs in analogous late Ediacaran settings globally—may be more likely to reflect the lack of animal activity on land during this period. Indeed, the absence of 'classic' Ediacaran macrofossils (including metazoans [63]) in the terrestrial deposits of the St John's group is well attested (e.g. [15,17]). The discovery of well-preserved macroscopic fossils (albeit of probable communal microbial origin) in at least two taphonomic modes (aluminosilicification and phosphatization) within this fluvioterrestrial succession suggests that this absence is not simply preservational, and supports the general consensus that complex macroorganisms, although well established in marine systems, had not yet colonized terrestrial settings.

The discovery of large streamers suggests that microbial and perhaps algal life was highly abundant in some terrestrial settings by the late Ediacaran. Indeed, the abundance of arumberiamorph mats alongside the streamers at Ferryland Head, and their ubiquity in similar rocks globally, suggest that late Ediacaran fluvio-tidal systems could accumulate substantial photosynthetic and/or chemotrophic biomass; the apparent absence of animals was not for want of a food source. We note that microbial communities can potentially increase sediment cohesion on channel-bounding flood plains [64] and thus also affect the preserved architecture of fluvio-tidal systems [65,66]. Future studies of late Ediacaran nutrient cycling, non-marine sedimentology and the rise of life on land should take these results into account.

# 4. Methods

Optical microscopy was undertaken with a Leica DM2700 P reflected/transmitted light polarizing microscope with DFC 420C camera and Leica Application Suite v. 4.00. For compositional analysis of carbon-coated, polished, uncovered thin sections, scanning electron microscopy was undertaken at the Analysis and Characterization facility of the Aberdeen Centre for Electron Microscopy, University of Aberdeen, using a Carl Zeiss GeminiSEM 300 VP equipped with an Oxford Instruments NanoAnalysis Xmax80 X-ray energy-dispersive spectrometer.

Data accessibility. All relevant data are included in the manuscript and supplement. Thin sections figured in this paper (FRY-10 and NL-3), and corresponding hand samples, have been deposited at the Cockburn Museum, the University of Edinburgh, under accession no. EUCM.0001.2021. No permits were required to sample at the localities studied in this work. The data are provided in the electronic supplementary material [67].

Authors' contributions. S.M.: Conceptualization, data curation, investigation, methodology, visualization, writing—original draft, writing—review and editing; J.J.M.: conceptualization, data curation, investigation, methodology, resources, visualization, writing—original draft, writing—review and editing; A.B.: data curation, investigation, methodology, visualization, writing—review and editing; J.S.: data curation, investigation, methodology, visualization. All authors gave final approval for publication and agreed to be held accountable for the work performed therein.

Competing interests. We declare we have no competing interests.

Funding. S.M. acknowledges support from the European Union's Horizon 2020 Research and Innovation Programme under Marie Skłodowska-Curie grant agreement no. 747877, and from the NASA Astrobiology Institute NNA13AA90A, *Foundations of Complex Life, Evolution, Preservation and Detection on Earth and Beyond*. J.J.M. recognized funding from Novas Consulting, through the Geological Society of London, as well as from the Government of Newfoundland and Labrador.

Acknowledgements. We thank B. Lynne and J. Poling for help in obtaining (*a*), (*c*) and (*d*) of figure 4. Discussions with D. McIlroy and P. Smith, and the comments of William McMahon (no relation to S.M.) and three anonymous reviewers, significantly improved this work. We thank the people of Ferryland for their continued support and hospitality. The early stages of this research benefited greatly from the guidance of M. D. Brasier.

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
