## [Peer Review File · Proceedings of the Royal Society B: Biological Sciences]

Review History

RSPB-2021-0249.R0 (Original submission)

Review form: Reviewer 1 (Gregory Retallack)

Recommendation

Major revision is needed (please make suggestions in comments)

Scientific importance: Is the manuscript an original and important contribution to its field?
Acceptable

General interest: Is the paper of sufficient general interest?
Good

Quality of the paper: Is the overall quality of the paper suitable?
Good

Is the length of the paper justified?
Yes

Should the paper be seen by a specialist statistical reviewer?
No

Do you have any concerns about statistical analyses in this paper? If so, please specify them explicitly in your report.

No

It is a condition of publication that authors make their supporting data, code and materials available - either as supplementary material or hosted in an external repository. Please rate, if applicable, the supporting data on the following criteria.

Is it accessible?

Yes

Is it clear?

Yes

Is it adequate?

Yes

Do you have any ethical concerns with this paper?

No

Comments to the Author

This paper is a worthwhile addition to the literature on recognizing Ediacaran non-marine paleoenvironments and paleosols using a variety of microbially-induced sedimentary structures. The observation of streamers is a useful addition to the literature. Streamers are however not entirely subaerial. When I have seen modern streamers, they were in dried out channel beds, and not on soil surfaces. Thus I think it should be published, but only after major revision, fixing six significant errors.

First, the author's seem to think Arumberia is a microbially-induced sedimentary structure (l. 108), when a recent study of the holotype material, including thin sections, showed that it was a discrete vendobiont (Retallack and Broz, 2020a,b). The material of this paper may be Neantia rhodonensis, also discussed in those papers. The ichnogenera Neantia, Rugalichnus, Eoclathrus and Rivularites are all available for such MISS, but the examples described are not Arumberia. Second, is the claim that absence of evidence of vendobionts is evidence of absence in these rocks (l.28). Stratigraphic levels of vendobionts in measured sections (Retallack, 2012, 2013, 2019, Retallack and Broz 2020a,b, Retallack et al., 2021) show that non-marine Ediacaran rocks preserve vendobionts on only a few levels, just as many Phanerozoic non-marine rocks preserve fossil plants on few levels.

Third is the claim that for the studied rocks that "no doubt these are Ediacaran" (l.64). The long stratigraphic distance above the Mistaken Point level is less relevant for this purpose than the relationship with the basal Cambrian golden spike in the Chapel Island Formation at Fortune Head, in a different fault block some distance away. That stratotype locality is also partly non-marine as recognized from paleosols there (Shahkarami et al., 2020). Is there any relevant fossil evidence for the age of the rocks studied?

Fourth, how can a paper published in 2016 be challenged by papers published in 2003 and 2007 (l.38)? Also how robust and final is the challenge considering the all the recent papers listed below. This is still a fun and evolving controversy in my opinion, and this manuscript is not relevant to habitats of vendobionts, because of the three points above.

Fifth, the streamer morphology does not indicate a lack of animals (l.317). Modern examples dry hard and are uneaten in ephemeral rivers. That is my experience and the illustrated examples of this manuscript show it well.

Sixth, the structures observed are unlikely to be algal (l.326), and are more likely cyanobacterial. Most freshwater algae are coccoidal and do not form streamers, but cyanobacteria such as Nostoc and Scytonema commonly have that morphology.

Retallack, G.J., 2012. Were Ediacaran siliciclastics of South Australia coastal or deep marine? *Sedimentology* 59, 1208-1236.

Retallack, G.J., 2013a, Ediacaran life on land. *Nature* 493, 89-92.

Retallack, G.J., 2019. Interflag sandstone laminae, a novel fluvial sedimentary structure with implication for Ediacaran paleoenvironments. *Sedimentary Geology* 379, 60-76.

Retallack, G.J., 2020, Boron paleosalinity proxy for deeply buried Paleozoic and Ediacaran fossils. *Palaeogeography Palaeoclimatology Palaeoecology* 540, 109536.

Retallack, G.J., and Broz, A, 2020a, Arumberia and other Ediacaran fossils from central Australia. *Historical Biology* 32, 1755281.

Retallack G.J., and Broz, A., 2020b, Ediacaran and Cambrian paleosols in central Australia. *Palaeogeography Palaeoclimatology Palaeoecology* 560, 110047

Retallack, G.J., Matthews, N., Master, S., Khangar, R. and Khan, M., 2021, Dickinsonia discovered in India and late Ediacaran biogeography. *Gondwana Research* 90, 165–170.

Shahkarami, S., Buatois, L.A., Mángano, M.G., Hagadorn, J.W. and Almond, J., 2020. The Ediacaran–Cambrian boundary: evaluating stratigraphic completeness and the Great Unconformity. *Precambrian Research*, 345, p.105721.

Minor points

l.100 “old elephant skin” often has healed desiccation cracks, so is not merely an undermat

Review form: Reviewer 2

Recommendation

Accept with minor revision (please list in comments)

Scientific importance: Is the manuscript an original and important contribution to its field?

Excellent

General interest: Is the paper of sufficient general interest?

Good

Quality of the paper: Is the overall quality of the paper suitable?

Good

Is the length of the paper justified?

Yes

Should the paper be seen by a specialist statistical reviewer?

No

Do you have any concerns about statistical analyses in this paper? If so, please specify them explicitly in your report.

No

It is a condition of publication that authors make their supporting data, code and materials available - either as supplementary material or hosted in an external repository. Please rate, if applicable, the supporting data on the following criteria.

Is it accessible?

Yes

Is it clear?

Yes

Is it adequate?

Yes

Do you have any ethical concerns with this paper?

No

Comments to the Author

I have enjoyed reading about these intriguing structures. The dessication features in the Gibbett Fill Formation look convincing. The structures from Ferryland Hill are more difficult. In particular rather hard to understand preservation in positive epirelief in argillaceous material but as mentioned in the manuscript similar preservation has been previously described. But could a cartoon be provided how the authors envisage steps in this preservation?

Apart from that my two main comments are on nomenclature related to *Arumberia* where I believe the authors need to argue this (in supplementary information). The second is on some rather washed-out images.

TAXONOMY AND NOMENCLATURE

I would suggest a second thought on the use of “pseudo-taxon” (line 147), which I quite frankly had not seen before. In the meaning of “pseudo” as something false or spurious do the authors mean that it is not a true or valid taxon? *Arumberia* as described by Glaessner & Walter (1975) certainly full-fills the formal criteria for the erection of a new taxon. Another question is if *Arumberia* is a fossil, a dubiofossil or a pseudofossil. A common convention for a taxon later found to be a pseudofossil is to maintain the name but not in italics. In the manuscript *Arumberia* is used without italics, which would be consistent with this, but if I read the manuscript correctly, the authors do see *Arumberia* as a fossil, not a pseudofossil (cf. line 66). A minority view (e.g. Stimpson et al. 2017: An ichnotaxonomic approach to wrinkled microbially induced sedimentary structures. *Ichnos* 24, 291-316, and further papers by lead author) is to consider taxa erected on MISS as ichnotaxa and so use in italics. The authors state (line 193) “We consider these megascopic ribbons to be probable fossils...”, and later (lines 256-58) “We agree with previous authors that ‘*Arumberia*’ represents an organic matground morphotype arising from the interaction of dense biotic communities, siliciclastic sediment, and flowing water.”. There seems to be some ambiguity here if the authors really see the streamers as related with *Arumberia*. Better use lower case “A” for “arumberiamorphs” (line 154, and throughout manuscript).

My suggestion from the above is that the authors present, probably best in the supplementary information, explanation on how they are using *Arumberia*. This would also make redundant the use of quotations marks.

FIGURES

Parts B, C and D in Figure 2 are rather washed-out and hard to make out detail. I assume these were images taken on a cloudy and murky day and that the authors do not have better contrast images, but could some of that be compensated for in the course of image processing?

Part A, Figure 2. At the scale given hard to see anything of these gypsum pseudomorphs. If the idea is to show any of the morphology rather than a surface with dots then a close-up should be used.

The above comment applies also to washed-out appearance of structures in Figure 3. There is no explanation in Figure caption as to significance of arrows.

OTHER

Line 16, “... these organisms ...” do not follow from preceding text.

Line 65, “... but there is no doubt these rocks are Ediacaran ...”. I have no reason to doubt this but the sentence seems to call out for a because, which could be addressed in the supplementary information.

Line 166, for clarity specify already here if on bed top or bed sole, information first provided on line 175. Also add this information to the caption to figure 4.

Line 168, should “most unlike” be “unlike most”?

As I understand it all figured material is still in the field. Storage information on the thin-sections is, I believe, missing.

Review form: Reviewer 3 (William McMahon)

Recommendation

Major revision is needed (please make suggestions in comments)

Scientific importance: Is the manuscript an original and important contribution to its field?

Good

General interest: Is the paper of sufficient general interest?

Good

Quality of the paper: Is the overall quality of the paper suitable?

Marginal

Is the length of the paper justified?

Yes

Should the paper be seen by a specialist statistical reviewer?

No

Do you have any concerns about statistical analyses in this paper? If so, please specify them explicitly in your report.

No

It is a condition of publication that authors make their supporting data, code and materials available - either as supplementary material or hosted in an external repository. Please rate, if applicable, the supporting data on the following criteria.

Is it accessible?

Yes

Is it clear?

Yes

Is it adequate?

Yes

Do you have any ethical concerns with this paper?

No

Comments to the Author

See attached document and photograph. (See Appendix A)

Decision letter (RSPB-2021-0249.R0)

02-Mar-2021

Dear Dr Matthews:

I am writing to inform you that your manuscript RSPB-2021-0249 entitled "Late Ediacaran life on land: desiccated microbial mats and fluvial biofilm streamers" has, in its current form, been rejected for publication in Proceedings B.

This action has been taken on the advice of referees, who have recommended that substantial revisions are necessary. With this in mind we would be happy to consider a resubmission, provided the comments of the referees are fully addressed. However please note that this is not a provisional acceptance.

Note that information on the storage of photomicrographs in Figure 4 will be necessary to comply with our data policies.

Sincerely,

Dr John Hutchinson, Editor

Associate Editor

Comments to Author:

Dear Authors

You will see that the reviewers all have some concerns and considerations for you to deliberate on. Views are altogether variable in terms of what they consider compelling or not and you will need to bring those together. In particular reviewer 3 has concerns about the sedimentological framework and whether this is as aerially exposed as claimed.

Reviewer(s)' Comments to Author:

Referee: 1

Comments to the Author(s)

This paper is a worthwhile addition to the literature on recognizing Ediacaran non-marine paleoenvironments and paleosols using a variety of microbially-induced sedimentary structures. The observation of streamers is a useful addition to the literature. Streamers are however not entirely subaerial. When I have seen modern streamers, they were in dried out channel beds, and not on soil surfaces. Thus I think it should be published, but only after major revision, fixing six significant errors.

First, the author's seem to think Arumberia is a microbially-induced sedimentary structure (l. 108), when a recent study of the holotype material, including thin sections, showed that it was a discrete vendobiont (Retallack and Broz, 2020a,b). The material of this paper may be Neantia rhedonensis, also discussed in those papers. The ichnogenera Neantia, Rugalichnus, Eoclathrus and Rivularites are all available for such MISS, but the examples described are not Arumberia. Second, is the claim that absence of evidence of vendobionts is evidence of absence in these rocks (l.28). Stratigraphic levels of vendobionts in measured sections (Retallack, 2012, 2013, 2019, Retallack and Broz 2020a,b, Retallack et al., 2021) show that non-marine Ediacaran rocks preserve vendobionts on only a few levels, just as many Phanerozoic non-marine rocks preserve fossil plants on few levels.

Third is the claim that for the studied rocks that "no doubt these are Ediacaran" (l.64). The long stratigraphic distance above the Mistaken Point level is less relevant for this purpose than the relationship with the basal Cambrian golden spike in the Chapel Island Formation at Fortune Head, in a different fault block some distance away. That stratotype locality is also partly non-marine as recognized from paleosols there (Shahkarami et al., 2020). Is there any relevant fossil evidence for the age of the rocks studied?

Fourth, how can a paper published in 2016 be challenged by papers published in 2003 and 2007 (l.38)? Also how robust and final is the challenge considering the all the recent papers listed below. This is still a fun and evolving controversy in my opinion, and this manuscript is not relevant to habitats of vendobionts, because of the three points above.

Fifth, the streamer morphology does not indicate a lack of animals (l.317). Modern examples dry hard and are uneaten in ephemeral rivers. That is my experience and the illustrated examples of this manuscript show it well.

Sixth, the structures observed are unlikely to be algal (l.326), and are more likely cyanobacterial. Most freshwater algae are coccoidal and do not form streamers, but cyanobacteria such as Nostoc and Scytonema commonly have that morphology.

Retallack, G.J., 2012. Were Ediacaran siliciclastics of South Australia coastal or deep marine? *Sedimentology* 59, 1208-1236.

Retallack, G.J., 2013a, Ediacaran life on land. *Nature* 493, 89-92.

Retallack, G.J., 2019. Interflag sandstone laminae, a novel fluvial sedimentary structure with implication for Ediacaran paleoenvironments. *Sedimentary Geology* 379, 60-76.

Retallack, G.J., 2020, Boron paleosalinity proxy for deeply buried Paleozoic and Ediacaran fossils. *Palaeogeography Palaeoclimatology Palaeoecology* 540, 109536.

Retallack, G.J., and Broz, A, 2020a, Arumberia and other Ediacaran fossils from central Australia. *Historical Biology* 32, 1755281.

Retallack G.J., and Broz, A., 2020b, Ediacaran and Cambrian paleosols in central Australia. *Palaeogeography Palaeoclimatology Palaeoecology* 560, 110047

Retallack, G.J., Matthews, N., Master, S., Khangar, R. and Khan, M., 2021, Dickinsonia discovered in India and late Ediacaran biogeography. *Gondwana Research* 90, 165-170.

Shahkarami, S., Buatois, L.A., Mángano, M.G., Hagadorn, J.W. and Almond, J., 2020. The Ediacaran-Cambrian boundary: evaluating stratigraphic completeness and the Great Unconformity. *Precambrian Research*, 345, p.105721.

Minor points

l.100 "old elephant skin" often has healed desiccation cracks, so is not merely an undermat

Referee: 2

Comments to the Author(s)

I have enjoyed reading about these intriguing structures. The dessication features in the Gibbett Fill Formation look convincing. The structures from Ferryland Hill are more difficult. In particular rather hard to understand preservation in positive epirelief in argillaceous material but as mentioned in the manuscript similar preservation has been previously described. But could a cartoon be provided how the authors envisage steps in this preservation?

Apart from that my two main comments are on nomenclature related to Arumberia where I believe the authors need to argue this (in supplementary information). The second is on some rather washed-out images.

TAXONOMY AND NOMENCLATURE

I would suggest a second thought on the use of “pseudo-taxon” (line 147), which I quite frankly had not seen before. In the meaning of “pseudo” as something false or spurious do the authors mean that it is not a true or valid taxon? Arumberia as described by Glaessner & Walter (1975) certainly full-fills the formal criteria for the erection of a new taxon. Another question is if Arumberia is a fossil, a dubiofossil or a pseudofossil. A common convention for a taxon later found to be a pseudofossil is to maintain the name but not in italics. In the manuscript Arumberia is used without italics, which would be consistent with this, but if I read the manuscript correctly, the authors do see Arumberia as a fossil, not a pseudofossil (cf. line 66). A minority view (e.g. Stimpson et al. 2017: An ichnotaxonomic approach to wrinkled microbially induced sedimentary structures. *Ichnos* 24, 291-316, and further papers by lead author) is to consider taxa erected on MISS as ichnotaxa and so use in italics. The authors state (line 193) “We consider these megascopic ribbons to be probable fossils...”, and later (lines 256-58) “We agree with previous authors that ‘Arumberia’ represents an organic matground morphotype arising from the interaction of dense biotic communities, siliciclastic sediment, and flowing water.”. There seems to be some ambiguity here if the authors really see the streamers as related with Arumberia. Better use lower case “A” for “arumberiamorphs” (line 154, and throughout manuscript). My suggestion from the above is that the authors present, probably best in the supplementary information, explanation on how they are using Arumberia. This would also make redundant the use of quotations marks.

FIGURES

Parts B, C and D in Figure 2 are rather washed-out and hard to make out detail. I assume these were images taken on a cloudy and murky day and that the authors do not have better contrast images, but could some of that be compensated for in the course of image processing?

Part A, Figure 2. At the scale given hard to see anything of these gypsum pseudomorphs. If the idea is to show any of the morphology rather than a surface with dots then a close-up should be used.

The above comment applies also to washed-out appearance of structures in Figure 3. There is no explanation in Figure caption as to significance of arrows.

OTHER

Line 16, “... these organisms ...” do not follow from preceding text.

Line 65, “... but there is no doubt these rocks are Ediacaran ...”. I have no reason to doubt this but the sentence seems to call out for a because ..., which could be addressed in the supplementary information.

Line 166, for clarity specify already here if on bed top or bed sole, information first provided on line 175. Also add this information to the caption to figure 4.

Line 168, should “most unlike” be “unlike most”?

As I understand it all figured material is still in the field. Storage information on the thin-sections is, I believe, missing.

Referee: 3

Comments to the Author(s)

See attached document and photograph

Author's Response to Decision Letter for (RSPB-2021-0249.R0)

See Appendix B.

RSPB-2021-1875.R0

Review form: Reviewer 1 (Gregory Retallack)

Recommendation

Accept with minor revision (please list in comments)

Scientific importance: Is the manuscript an original and important contribution to its field?

Good

General interest: Is the paper of sufficient general interest?

Excellent

Quality of the paper: Is the overall quality of the paper suitable?

Good

Is the length of the paper justified?

Yes

Should the paper be seen by a specialist statistical reviewer?

No

Do you have any concerns about statistical analyses in this paper? If so, please specify them explicitly in your report.

No

It is a condition of publication that authors make their supporting data, code and materials available - either as supplementary material or hosted in an external repository. Please rate, if applicable, the supporting data on the following criteria.

Is it accessible?

Yes

Is it clear?

Yes

Is it adequate?

No

Do you have any ethical concerns with this paper?

No

Comments to the Author

This is improved however there is one issue and one minor glitch still

The main problem is that Arumberia is not universally regarded as a mat structure. The following paper should be cited and reasons given why interpretation of Arumberia as a discrete fossil is incorrect.

Retallack, G.J., and Broz, A, 2020, Arumberia and other Ediacaran fossils from central Australia. *Historical Biology* 32, 1755281.

Why is the title to Tarhan et al. (2017 in all caps)

Review form: Reviewer 3 (William McMahon)

Recommendation

Accept with minor revision (please list in comments)

Scientific importance: Is the manuscript an original and important contribution to its field?

Good

General interest: Is the paper of sufficient general interest?

Good

Quality of the paper: Is the overall quality of the paper suitable?

Good

Is the length of the paper justified?

Yes

Should the paper be seen by a specialist statistical reviewer?

No

Do you have any concerns about statistical analyses in this paper? If so, please specify them explicitly in your report.

No

It is a condition of publication that authors make their supporting data, code and materials available - either as supplementary material or hosted in an external repository. Please rate, if applicable, the supporting data on the following criteria.

Is it accessible?

Yes

Is it clear?

Yes

Is it adequate?

Yes

Do you have any ethical concerns with this paper?

No

Comments to the Author

See attached file. (See Appendix C)

Review form: Reviewer 4**Recommendation**

Accept with minor revision (please list in comments)

Scientific importance: Is the manuscript an original and important contribution to its field?

Good

General interest: Is the paper of sufficient general interest?

Excellent

Quality of the paper: Is the overall quality of the paper suitable?

Good

Is the length of the paper justified?

Yes

Should the paper be seen by a specialist statistical reviewer?

No

Do you have any concerns about statistical analyses in this paper? If so, please specify them explicitly in your report.

No

It is a condition of publication that authors make their supporting data, code and materials available - either as supplementary material or hosted in an external repository. Please rate, if applicable, the supporting data on the following criteria.

Is it accessible?

Yes

Is it clear?

Yes

Is it adequate?

Yes

Do you have any ethical concerns with this paper?

No

Comments to the Author

McMahon et al. present new fossils from Ediacaran strata in Newfoundland, Canada, that provide evidence of microbial life in emergent/terrestrial settings. The timing of life's emergence onto the land surface is controversial. Geochemical evidence points to microbial life on land going back far into Precambrian time, but the fossil record is sparse at best. The evidence presented by McMahon et al. then is exciting, particularly since it provides the earliest evidence of an ecological mode of microbial life, "streamers", that is common today.

The fossil evidence presented is of high quality and I have relatively few comments. A major point of concern for me, the geological setting of the fossils, has already been addressed by the authors in response to other reviewers' comments. In particular, the latest revision of the manuscript provides a less specific interpretation of the depositional environment, which I think is supported by the available data (although I agree with reviewer 3 that future work should concentrate on further data collection to unravel the precise palaeoenvironmental context of these fossils). The broader interpretation of palaeoenvironment in this version of the manuscript does not diminish the significance of the finding in my opinion.

I think this paper should be published in Proceedings B. I enclose a pdf with minor comments. Diverging from the other reviewers, my two substantive remaining comments refer to (i) the interpretation that these fossils are mineralised in aluminosilicates and the evidence presented to support this, and (ii) the organisation of the Results/Discussion.

Mineralised in aluminosilicates

The authors argue in lines 219–234 that the ribbons have been replaced by clay minerals, but they show little evidence beyond a single thin-section to support this assertion. Are there any geochemical data to support this claim e.g., EDS? Has all the carbon completely gone? Given the context of bacterial-clay interactions citing some of the work of Kurt Konhauser and colleagues seems appropriate.

Organisation of the Results/Discussion

I think there should be more delineation between the results and discussion sections of the manuscript. The final section of the results conflates reporting/interpretation of the streamers only (lines 167–254) with a discussion of how they might be grouped on a continuum with *Arumberia*. I think line 256 would mark a convenient point to begin a new section that deals exclusively with the latter point, reducing confusion for the reader. The paper would then report the three types of structure/fossils seen, discuss how two of them may lie on a continuum, before discussing the significance of the trio.

Decision letter (RSPB-2021-1875.R0)

16-Sep-2021

Dear Dr Matthews:

Your manuscript has now been peer reviewed and the reviews have been assessed by an Associate Editor. The reviewers' comments (not including confidential comments to the Editor) and the comments from the Associate Editor are included at the end of this email for your reference. As you will see, the reviewers and the Editors have raised some concerns with your manuscript and we would like to invite you to revise your manuscript to address them. Critiques have been quite constructive and the paper has improved markedly; well done.

Research ethics:

Use of animals and field studies:

It is a condition of publication that you make available the data and research materials supporting the results in the article (<https://royalsociety.org/journals/authors/author-guidelines/#data>). Datasets should be deposited in an appropriate publicly available repository and details of the associated accession number, link or DOI to the datasets must be included in the Data Accessibility section of the article (<https://royalsociety.org/journals/ethics-policies/data-sharing-mining/>). Reference(s) to datasets should also be included in the reference list of the article with DOIs (where available).

Please submit a copy of your revised paper within three weeks. If we do not hear from you within this time your manuscript will be rejected. If you are unable to meet this deadline please let us know as soon as possible, as we may be able to grant a short extension.

Best wishes,
Dr John Hutchinson, Editor
mailto:proceedingsb@royalsociety.org

Associate Editor Board Member
Comments to Author:
Dear Authors

You will see that the reviewers are happy with your revisions and have a few additional comments. An additional review from a fourth reviewer lists some additional suggestions that you should consider.

With respect to aluminosilicification, as is addressed here, I would also like to highlight the recent publication by Becker Kerber, that advocate a secondary replacement of metamorphosed organic material by aluminosilicates. I should also stress in relation to the study by Locatelli et al. (2017) that in my own experience looking at terrestrial soft tissue fossils where the rocks evidently have experienced some weathering due to flow of meteoric water you also see films replacing the organic material partly by clay minerals and partly by iron oxides.

My suggestion would be to perhaps either address why the observed microbial mats should have been preserved by primary aluminosilicification, or let it be open to having been incurred during later stage volatilisation of organically preserved microbial mat and infilling with aluminosilicates.

Sedimentological observations may allow to distinguish between these two by analysing the clay mineral fabric under secondary electron microscopy/back scattered electrons? However, it may not be of primary interest for the nature of the current study in which case reverting to simply acknowledging the potential primary and secondary mechanisms for the observed phenomenon would be a simpler solution.

Becker-Kerber, B., Abd Elmola, A., Zhuravlev, A., Gaucher, C., Simões, M.G., Prado, G.M.E.M., Vintaned, J.G., Fontaine, C., Lino, L.M., Sanchez, D.F. and Galante, D., 2021. Clay templates in Ediacaran vendotaeniaceans: Implications for the taphonomy of carbonaceous fossils. *GSA Bulletin*.

Reviewer(s)' Comments to Author:
Referee: 3
Comments to the Author(s).
See attached file.

Referee: 1
Comments to the Author(s).
This is improved however there is one issue and one minor glitch still

The main problem is that Arumberia is not universally regarded as a mat structure. The following paper should be cited and reasons given why interpretation of Arumberia as a discrete fossil is incorrect.

Retallack, G.J., and Broz, A, 2020, Arumberia and other Ediacaran fossils from central Australia. *Historical Biology* 32, 1755281.

Why is the title to Tarhan et al. (2017 in all caps?

Referee: 4

Comments to the Author(s).

McMahon et al. present new fossils from Ediacaran strata in Newfoundland, Canada, that provide evidence of microbial life in emergent/terrestrial settings. The timing of life's emergence onto the land surface is controversial. Geochemical evidence points to microbial life on land going back far into Precambrian time, but the fossil record is sparse at best. The evidence presented by McMahon et al. then is exciting, particularly since it provides the earliest evidence of an ecological mode of microbial life, "streamers", that is common today.

The fossil evidence presented is of high quality and I have relatively few comments. A major point of concern for me, the geological setting of the fossils, has already been addressed by the authors in response to other reviewers' comments. In particular, the latest revision of the manuscript provides a less specific interpretation of the depositional environment, which I think is supported by the available data (although I agree with reviewer 3 that future work should concentrate on further data collection to unravel the precise palaeoenvironmental context of these fossils). The broader interpretation of palaeoenvironment in this version of the manuscript does not diminish the significance of the finding in my opinion.

I think this paper should be published in Proceedings B. I enclose a pdf with minor comments. Diverging from the other reviewers, my two substantive remaining comments refer to (i) the interpretation that these fossils are mineralised in aluminosilicates and the evidence presented to support this, and (ii) the organisation of the Results/Discussion.

Mineralised in aluminosilicates

The authors argue in lines 219–234 that the ribbons have been replaced by clay minerals, but they show little evidence beyond a single thin-section to support this assertion. Are there any geochemical data to support this claim e.g., EDS? Has all the carbon completely gone? Given the context of bacterial-clay interactions citing some of the work of Kurt Konhauser and colleagues seems appropriate.

Organisation of the Results/Discussion

I think there should be more delineation between the results and discussion sections of the manuscript. The final section of the results conflates reporting/interpretation of the streamers only (lines 167–254) with a discussion of how they might be grouped on a continuum with Arumberia. I think line 256 would mark a convenient point to begin a new section that deals exclusively with the latter point, reducing confusion for the reader. The paper would then report the three types of structure/fossils seen, discuss how two of them may lie on a continuum, before discussing the significance of the trio.

Author's Response to Decision Letter for (RSPB-2021-1875.R0)

See Appendix D.

Decision letter (RSPB-2021-1875.R1)

14-Oct-2021

Dear Dr Matthews

I am pleased to inform you that your manuscript entitled "Late Ediacaran life on land: desiccated microbial mats and large biofilm streamers" has been accepted for publication in Proceedings B. Congratulations!!

Data Accessibility section

Open Access

You are invited to opt for Open Access, making your freely available to all as soon as it is ready for publication under a CCBY licence. Our article processing charge for Open Access is £1700. Corresponding authors from member institutions (<http://royalsocietypublishing.org/site/librarians/allmembers.xhtml>) receive a 25% discount to these charges. For more information please visit <http://royalsocietypublishing.org/open-access>.

Paper charges

Sincerely,

Dr John Hutchinson

Associate Editor:

Comments to Author:

Dear Authors

With the thoughtful considerations made to the reviewers I think this has stimulated enough considerations for the manuscript to have developed into a publication ready version.

I am happy to let it move to the next stage.

Yours

Jakob Vinther

Appendix A

Late Ediacaran life on land: desiccated microbial mats and fluvial biofilm streamers

McMahon et al.

McMahon et al. describe “macroscopic biofilm streamers” from the Ediacaran Ferryland Head Formation in Newfoundland, Canada. In addition to these really intriguing features, they also figure beautiful desiccated microbial mats and another occurrence of *Arumberia*, which is becoming an increasingly well-known surface texture from Late Neoproterozoic to lower Cambrian red beds. The descriptions of the morphology of the streamers and other surface textures are detailed and seemingly robust and will be of wide interest to readers of RSPB. Particularly important is that these structures were observed on emergent bedding planes, thus the observations further our understanding of known microbial ecosystems which had colonized the continents at this point in Earth history. I am not an expert on the occurrence of biofilm streamers, so have restricted my comments largely to the descriptive sedimentology and the inferred environments in which the streamers and other distinctive surface textures occupied.

I would like to see the manuscript published in RSPB, but first I feel the sedimentological framework under which these significant observations were made needs substantial improvement. Whilst I agree that the studied bedding plane of interest formed subaerially (probably as a tidally-influenced alluvial plain), I think the evidence presented for it accumulating under freshwater conditions is underwhelming. This is not to say the finding is not significant – it is – added complexity to the range of microbial ecosystems that we already know existed in continental environments at this period in Earth history. Only to say that I think there is robust sedimentological evidence suggestive of the key bedding planes accumulating in a coastal/tidally-influenced (periodically emergent) setting unlikely to be dominated by freshwater. There is actually very little sedimentological detail presented of any sort (a c. 9 m graphic log), with the reader asked to rely on a previous unpublished master’s thesis (Sala Toledo 2004) and geological notes accompanying a regional map (Williams and King, 1979) (Line 126) to confirm that their interpretation of a freshwater microbial ecosystem is robust. This is an opportunity to detail the sedimentology of an important geological succession to a wider audience and the authors must utilize it. Clearly length of the article is an issue here, but descriptions and interpretations of each sedimentary facies with accompanying figures should be added at least to the supplementary materials. The level of descriptive detail provided for the streamer morphology is superb, but this must be matched by a similar level of detail concerning the host sedimentary rocks.

I have split my comments into 2 “major points” (“The earliest record of an important mode of life for freshwater microbial ecosystems” and “Order of results”) as well as a series of line-by-line comments (which range from minor to quite significant). I hope this work will assist the authors in their aim of publishing details of these great bedding planes in RSPB, but I hope they also realise that first a significant amount of sedimentological detail must be added, and perhaps some revisions to the present interpretation. I emphasize that the finding is still very important and worthy of publication in RSPB even if not an unequivocal fluvial microbial ecosystem.

“The earliest record of an important mode of life for freshwater microbial ecosystems”.

I find two things very concerning with this key statement: 1) evidence for a **freshwater** depositional environment; and 2) the claimed significance of the find.

1) *Evidence for freshwater*: The amount of original sedimentology presented from the succession to demonstrate the interpreted depositional environment is a little disappointing. Only a c. 9 m stratigraphic log is presented, with no accompanying photographs and no dedicated section (in the supplementary or main text) devoted to the description of each facies and interpretation. In my opinion this is insufficient in order to prove the proposed freshwater depositional environment. A recent publication in RSPB detailed non-marine (brackish-water) trilobites (<https://doi.org/10.1098/rspb.2020.2263>) in which the study included multiple

sedimentary logs in their supplementary information, as well as 2 panels of photographs illustrating key facies and a very detailed table of facies. I would expect a similar level of sedimentological detail more than achievable from the Ferryland Head outcrops.

At present, this study contains one small sedimentary log and a sedimentology section spanning 9 lines (Line 116-124). There is certainly opportunity to expand this by making use of the supplementary material. I do not think the buff-coloured sandstones are arkosic but arenitic. This is a small detail though, more important is that these sandstones almost always contain flaser cross-stratification (or at least mud-draped ripples) – see attached photograph. The red-brown siltstone rich facies encase the channelized buff-coloured sandstone demonstrating that these two facies were not only coevally occurring but represent laterally adjacent settings. The buff-coloured sandstones are evidently tidal (mud-draped laminae, also bidirectional paleocurrents, sandstone bodies show evident split into mutually-evasive ebb and flood channels), so in my opinion a freshwater origin for the red-brown siltstones is extremely unlikely. Note also that (depending on channel-bed slope and tidal range), fluvial-tidal transitions may extend tens of hundreds of kilometres inland from river mouths – really difficult to prove that the red-brown siltstone facies was freshwater even if treated separately from the buff-coloured sandstones.

The authors state on Line 120 that their c. 10 m (actually closer to 9 m) section is dominated by the red-brown siltstone rich facies: I can pass a couple more comments on the descriptive sedimentology and possible interpretation of this facies which I hope are useful for the authors when making their revision. Note importantly though that it is the association between the red-brown siltstones and the clearly tidal buff-coloured sandstone facies is key to its interpretation (another reason why the single 9 m thick log would ideally be expanded).

- The red-coloured, fine to very fine-grained, predominantly parallel-laminated silty sandstones display current ripple cross-lamination and soft sediment deformation. They occur in association with channelized (“buff-coloured”) sandstones predominantly filled with mud-draped ripple cross-laminae. Also intercalated are red-coloured mudstones which host *Arumberia* and the described “streamers”. Intercalated sandstones show current- and wave-formed ripple lamination. Desiccated polygons are also widespread, in addition to other surface textures showing evidence of emergence (e.g., adhesion marks, gypsum pseudomorphs).
- I suspect the red-brown siltstone rich facies preserves a tidally-influenced alluvial plain. Periodic emergence is supported by the widespread presence of desiccation cracks, whereas other surface textures indicate periodic submergence (e.g., wave-formed ripple lamination). Whereas the buff-coloured sandstones likely formed in tidally-influenced fluvial channels/tidal channels/estuarine channels, the red silty sandstones represent unconfined deposition. Water escape structures suggest periodic saturation (fluctuating water table?). The red mudstones which preserve *Arumberia*/streamers represent deposition on a tidally-influenced alluvial plain (or a playa/mudflat?). The distinction between these would be that alluvial plains are fed by river systems (perhaps the buff-coloured sandstone facies?). Association with tidal sedimentary structures goes against a freshwater origin for the streamer-hosting bed. Whilst the beds are definitely continental/non-marine – stating that they are freshwater, and therefore that the observed streamers preserve a freshwater microbial ecosystem, is a step too far (and I don’t think valid). This becomes very significant in the discussion, where the term “fluvial” is used repeatedly (and I think incorrectly).

The authors are free to disagree with my opinion on the unit’s sedimentology, but if they want to press their case for a freshwater origin and “fluvial streamers” then substantially more sedimentological detail is required. Personally, I think the term “freshwater” should simply be disregarded in all instances – and replaced with “continental” or “terrestrial”. This would still

be a significant find. As a minimum: description and interpretation for each facies in the Ferryland Head Formation is required – not simply asking the reader to refer to two previous reports (Line 126), one a report accompanying a geological map and the other a master’s thesis – both very hard to track down! The extent of the facies description of the Ferryland Head Formation in the former of these is 6 lines – and hardly providing a compelling case of deposition “in fluvial conditions” (Line 127). The original authors in that study do not describe the formation as representing fluvial conditions but forming beds “that appear to be channel sand deposits” (so why not tidal/estuarine/tidally-influenced etc. Not necessarily fluvial). Ideally a log would also be presented throughout the entire exposed section – this doesn’t necessarily need to be as high resolution as the 9 m stratigraphic log presented, which is the key section undoubtedly, but it should be there. Perhaps there is one already available in the Matthews 2011 thesis?

2) *Significance of find*: The paper frames the find as a significant “earliest record of an important mode of life for freshwater microbial ecosystems”. There is no clarification on what this “mode of life” actually is? Do the authors specifically mean “streamers”: as in, this is the earliest “fluvial biofilm streamers” reported? Or do they mean any form of freshwater microbial life. Notwithstanding my above concerns of the studied bedding plane hosting the streamers as having a freshwater origin, if the latter is intended then there are a number of outstanding citations of earlier reports. For example, microbially-induced sedimentary structures from the more unequivocally freshwater Mesoproterozoic Meall Dearg Formation in Scotland (Prave, 2002, [https://doi.org/10.1130/0091-7613\(2002\)030%3C0811:LOLITP%3E2.0.CO;2](https://doi.org/10.1130/0091-7613(2002)030%3C0811:LOLITP%3E2.0.CO;2) - see also McMahon and Davies, 2018, <https://doi.org/10.1144/jgs2017-012> - though this second study is far less important). Also, fluvio-lacustrine stromatolites from the Mesoproterozoic Copper Harbour Formation (Fedorchuk et al. 2016 - <https://doi.org/10.1016/j.precamres.2016.01.015>). There are three additional MISS reports from earlier strata reported in the cited Davies et al. (2016, ESR) work, their Table 1 – though I cannot comment on the veracity of any of these claims.

Order of results

I am curious as to why the authors have decided to present the desiccated microbial mats with “flip-overs” first, followed by *Arumberia*, followed by the macroscopic biofilm streamers. It seems to me that the last of these is by far the most significant finding – if indeed the authors are able to prove that these biofilms did occupy **freshwater fluvial** environments (and even if not, I think it is still the most noteworthy find). There have been a number of MISS reports from tidal environments earlier than this study (see Table 1 in the cited ESR paper by Davies et al.) and no less than 39 *Arumberia* case studies in total. What’s truly novel about this study is the report of the streamers. That’s not to say that the *Arumberia* and MISS reports aren’t interesting – they are – but I would have thought secondary.

Of course, if the authors can demonstrate why they chose to present the results in the order they did (stratigraphic?) of course that is OK, just seemed peculiar whilst reading.

Line-by-line:

Line 17 (and elsewhere). “Fluvially-influenced”: What do the authors mean by this term? To my understanding it has no grounding in the literature until this point. If used it would have a significantly wide reach surely? Floodplains are fluvially-influenced, as well as the main channel and all secondary conduits, crevasse splays etc. Are oceans and lakes also “fluvially-influenced” if the rivers feed them? Perhaps the authors mean that tidal/marine processes are the dominant depositional process, with only a slight involvement of the feeding fluvial system influencing the final preservation? I suspect this is not the case though if the authors do believe the studied bedding plane formed in freshwater.

Line 40. Not intending to push my own work, but McMahon et al. (<https://doi.org/10.2110/jsr.2020.029>) contains substantially more details on the marine fossiliferous environments in comparison to the studies presently listed. Matthews et al. (2020) is a great study, but my understanding is that it provides information on chronostratigraphy, not a detailed sedimentological account of the prevailing depositional environments. Could also consider citing good papers on both the White Sea Biota in Russia (e.g., Grazhdankin, D.V., 2003. Structure and depositional environment of the Vendian Complex in the southeastern White Sea area. *Stratigraphy and Geological Correlation*, 11(4), pp.313-331) and Nama Biota in Namibia (e.g., Maloney, K.M., Boag, T.H., Facciol, A.J., Gibson, B.M., Cribb, A., Koester, B.E., Kenchington, C.G., Racicot, R.A., Darroch, S.A. and Laflamme, M., 2020. Paleoenvironmental analysis of Ernietta-bearing Ediacaran deposits in southern Namibia. *Palaeogeography, Palaeoclimatology, Palaeoecology*, 556, p.109884).

Line 42. Is this true? Davies et al. 2016 go to great lengths to highlight MISS distribution throughout the Precambrian and Phanerozoic (Section 2) and discuss how any increased MISS in the Ediacaran and Cambrian may be in part reflect a sampling bias due to greater interest in surface textures from this time period in comparison to, for example, the Silurian. It would be appropriate to change the sentence to “are, however, **commonly reported** in Late Ediacaran”. Note also that Table 1 contains the fluvial MISS reported by Prave (2002) from the Mesoproterozoic in addition to three other earlier MISS reports (all Mesoproterozoic). Kolesnikov (et al. 2017) is also a very curious reference to add in at this point. The paper discusses arumberia-like surface textures found on a modern saltern and make some comparisons between those and *Arumberia* in two Russian formations – neither of which have been described as occurring in “fluvially-influenced” depositional environments.

Line 53. Is there a citation that could be used for the statement here about the depositional environments of the Caphayden and Gibbett Hill formations?

Line 57. I would consider substituting “**fluvially-influenced**” to “**records the onset of periodically emergent conditions**”. This is far more accurate. Note also that marine red beds can also occur; for example, the Caerfai Bay Shales in Wales.

Line 81/Figure 1A. Beautiful!

Line 101. If it’s well known could we perhaps have a couple of more references?

Line 126. An unpublished thesis and a paragraph in a mapping report can hardly be considered a robust summary, given the significance of the claimed discovery. Additional sedimentological details should be presented here – or at the very least in the supplementary materials. See major comments.

Line 127. “We consider the laterally extensive bed geometries and observed sedimentary structures to be indicative of deposition within a braided river system”. Meandering channels can have extensive bed geometries as well? As for the observed sedimentary structures – very few have been presented?

Line 129. “The buff sandstone facies is here considered to be associated with proximal over-bank and crevasse-splay deposits, and transitions into the finer- grained facies”. I don’t agree with this. The buff-coloured sandstones form low-relief channels, contain mud-draped ripple cross laminae, as well as subordinate dune cross-strata. These are in my mind shallow and wide tidally-influenced channels. In order to show deposition occurring as crevasse-splays considerably more detail is required – a figure showing key faces, a photograph showing the extensive sand-body architecture etc. <https://doi.org/10.1016/j.sedgeo.2017.02.003> - This is a very good paper on crevasse-splay deposition that lays out robust criteria on recognizing such styles of deposition in ancient stratigraphy.

Line 131. The red-brown facies encase the buff-coloured sandstones – and are not just “overlying” it.

Line 132. Which channels? This interpretation of the facies relies on only the overbanks being preserved throughout the entire exposed section? Did the authors observe channels anywhere in the c. 250 metre thick Ferryland Head Formation? I would suggest that the red-brown facies represents the “overbanks/floodplains/alluvial plains”, with the buff-coloured sandstones representing the channels?

Line 132. “periodic wetting and drying cycles”. What drives these cycles? Again – the observation fits well within a tidal setting. Wetting and drying driven as current drains water away following an interval of elevated water level (e.g., in the receding waters of a tidal prism). I’m also confused as to why the red-brown facies has been introduced here briefly, before a switch is made back to the buff-coloured sandstones, only for the red-brown facies to be introduced properly in the next paragraph?

Line 134. I don’t follow this logic at all. 1) “Wave ripples are consistent with this facies” – What does that mean? Wave ripples occur within this facies? Whilst it is possible that the wave ripples formed in ephemeral braidplain lakes agitated by wind, there are copious other environments in which such structures could form. I can’t help but feel that this reference was cotton-picked because it provided one mechanism which fitted the envisioned freshwater origin.

Line 135-137. Current ripples can form in a whole host of depositional environments. How do mud-draped current ripples (see attached photograph) develop in an unconfined crevasse splay?

Line 138. I don’t think it is necessary to make a link to the floodplain fines element of a geological textbook here.

Line 139-141. This seems random? Why does the reader need to know this statement about meandering river facies?

Line 141. Please provide some photographs of the key characteristics of the buff-coloured sandstone facies.

Line 149. I don’t think *Arumberia* has been definitively proven to occur in fluvial facies

Line 157. Could the authors pass comment on whether or not they think splitting *Arumberia* down to the species level is appropriate? Or whether they think different forms may reflect different burial conditions/moments of interment? Figure 6 does a really nice job, and suggests to me that the authors believe it is purely to do with the prevailing current/sedimentological conditions.

Figure 3. Really great bedding plane and nicely presented figure.

Line 195. Earlier the biogenic *Arumberia* was described as being equidistant. Don’t think anyone would believe these are ripples – think sentence can be omitted. Though I appreciate the caution.

Line 232. Are these streamers the same as the structures described as *Arumberia Beckeri* in Kolesnikov et al., 2012? (their figure 3).

Line 283 and 289. “Alluvial” probably more appropriate than “Fluvial” in both instances. I don’t think compelling evidence can be provided to demonstrate that the succession is fluvial in origin.

Line 304. Also *Arumberia Beckeri*? See above.

Line 309. Alluvial, not fluvial.

Line 309. I think a sentence is needed somewhere to demonstrate that these were not wholly non-marine environments, but periodically emergent environments. Life was clearly able to take hold and survive subaerially – that’s the significance.

Line 312. Do coastal examples also exist?

Line 321. Fluvio-terrestrial – Why introduce this term at this point in the manuscript?

Line 327. I really think this should read some “subaerial” settings, or some “emergent” settings. I don’t think a fluvial interpretation will stand the test of time.

Line 329. Fluvio-tidal – another new term, but probably the most accurate used so far to describe the Ferryland Head succession.

Line 331-333. Is it necessary to note this? Note, McMahon et al. 2017 (already cited) state that the fluvial rock record will always average to the maximum normal deposition, and studies of pre-vegetation fluvial stratigraphy by and large have shown that this threshold exceeds the erosive threshold of surficial biofilms. Any eco-engineering capacity they provide daily is not archived in the ancient sedimentary rock record (it is lost when the biofilms ultimately fail). That said, present work in press in another journal have detailed how in pre-vegetation, low-gradient, muddy, coastal systems, engineering effects caused by microbial communities can have a greater chance of preservation. The cited study by Ielpi et al. 2018 only speculate on the impact of microbial life on alluvial architecture (there Section 5.4 “outstanding questions”). Another study perhaps of use is this recent work in ESPL, which concerns the role of microbiota in a modern coastal system: <https://doi.org/10.25850/nioz/7b.b.m>.

I have not checked the reference list.

Finally, I believe the conceptual model of the relationship between the streamers and more conventional *Arumberia* is superb and very valuable to the scientific community. I emphasize that I hope my criticisms are taken as constructive. With a sounder sedimentological framework, I believe the study will make a strong contribution to RSPB.

The authors are welcome to contact me if they have any questions relating to this review.

William McMahon

Appendix B

Note: Referee comments in Roman type, response/revisions in **bold**. Line numbers refer to those indicated in the left margin of the **tracked-changes version** of the revised manuscript. These line numbers show some discontinuities at page breaks because of a bug in MS Word.

Referee 1

Comments to the Author(s)

This paper is a worthwhile addition to the literature on recognizing Ediacaran non-marine paleoenvironments and paleosols using a variety of microbially-induced sedimentary structures. The observation of streamers is a useful addition to the literature. Streamers are however not entirely subaerial. When I have seen modern streamers, they were in dried out channel beds, and not on soil surfaces. Thus I think it should be published, but only after major revision, fixing six significant errors.

We agree that the streamers are not subaerial and do not anywhere claim otherwise; we describe in detail the role of water in their formation. These sediments are not palaeosols.

First, the author's seem to think Arumberia is a microbially-induced sedimentary structure (l. 108), when a recent study of the holotype material, including thin sections, showed that it was a discrete vendobiont (Retallack and Broz, 2020a,b). The material of this paper may be *Neantia rhedonensis*, also discussed in those papers. The ichnogenera *Neantia*, *Rugalichnus*, *Eoclathrus* and *Rivularites* are all available for such MISS, but the examples described are not *Arumberia*.

We disagree with the cited papers. The interpretation of Arumberia as a "vendobiont" is a fringe view with little support in the community because it is not adequately supported by the evidence so far presented. We would prefer not to enter this debate in the present manuscript but, briefly, we would argue that the interpretative diagrams in Retallack and Broz 2020a are speculative and contain numerous anatomical details not visible in the corresponding fossil material. Retallack and Broz 2020b do not provide any additional evidence in support of the "vendobiont" interpretation.

Second, is the claim that absence of evidence of vendobionts is evidence of absence in these rocks (l.28). Stratigraphic levels of vendobionts in measured sections (Retallack, 2012, 2013, 2019, Retallack and Broz 2020a,b, Retallack et al., 2021) show that non-marine Ediacaran rocks preserve vendobionts on only a few levels, just as many Phanerozoic non-marine rocks preserve fossil plants on few levels.

The key point here is that preservational conditions were evidently suitable for soft tissue preservation, since we have evidence of early diagenetic

aluminosilicification (replacing microbial filaments and streamers before they could decay) and phosphatization as well, both of which are well-known and important modes of metazoan soft tissue preservation in the Ediacaran. Thus, an absence of metazoans here cannot be attributed to unfavourable conditions for preservation. The simplest remaining explanation is that they were absent from the studied horizons. We do not claim to have searched the entire Formation but do note (with citations) that others have remarked on the absence of metazoan fossils from it. This comment did not prompt a revision.

Third is the claim that for the studied rocks that “no doubt these are Ediacaran” (l.64). The long stratigraphic distance above the Mistaken Point level is less relevant for this purpose than the relationship with the basal Cambrian golden spike in the Chapel Island Formation at Fortune Head, in a different fault block some distance away. That stratotype locality is also partly non-marine as recognized from paleosols there (Shahkarami et al., 2020). Is there any relevant fossil evidence for the age of the rocks studied?

This comment motivated clarifications on lines 85-90. The studied rocks are stratigraphically several km below the top of the Signal Hill Group, the upper part of which does not crop out in the area but correlates with the Musgravetown Group, which is itself overlain (unconformably) by the Lower Cambrian. There are no Phanerozoic fauna in either the Signal Hill or Musgravetown Groups, in contrast to the overlying Lower Cambrian Random Formation. Thus there is no doubt that the studied section is Precambrian. (These rocks do not directly correlate lithostratigraphically with the rocks of the Burin Peninsula such as the Chapel Island Formation).

Fourth, how can a paper published in 2016 be challenged by papers published in 2003 and 2007 (l.38)?

The reviewer makes a good point — this was an unfortunate choice of words, which we have corrected (instead of “has been robustly challenged, with...”, we now write, “conflicts with”).

Also how robust and final is the challenge considering the all the recent papers listed below. This is still a fun and evolving controversy in my opinion, and this manuscript is not relevant to habitats of vendobionts, because of the three points above.

We disagree for the reasons noted above.

Fifth, the streamer morphology does not indicate a lack of animals (l.317). Modern examples dry hard and are uneaten in ephemeral rivers. That is my experience and the illustrated examples of this manuscript show it well.

An association between abundant streamers and suppressed or absent animal activity is supported (for modern analogues) by the citations in the previous two sentences. However, we have moderated the wording here slightly to avoid undue confidence (from “more likely reflects” to “may be more likely to reflect”). The interpretation is also supported more straightforwardly by the lack of any animal fossils (or trace fossils), as we point out in the following sentence.

Sixth, the structures observed are unlikely to be algal (I.326), and are more likely cyanobacterial. Most freshwater algae are coccoidal and do not form streamers, but cyanobacteria such as *Nostoc* and *Scytonema* commonly have that morphology.

Unfortunately, there is not enough evidence to say whether the streamers at Ferryland Head were algal or cyanobacterial. The microscopic filaments associated with our streamers are large enough that they could have been eukaryotic, although it would not be surprising either if they were cyanobacterial. At the present day, streamers in non-extreme environments are commonly composed largely of eukaryotic algae.

Retallack, G.J., 2012. Were Ediacaran siliciclastics of South Australia coastal or deep marine? *Sedimentology* 59, 1208-1236.

Retallack, G.J., 2013a, Ediacaran life on land. *Nature* 493, 89-92.

Retallack, G.J., 2019. Interflag sandstone laminae, a novel fluvial sedimentary structure with implication for Ediacaran paleoenvironments. *Sedimentary Geology* 379, 60-76.

Retallack, G.J., 2020, Boron paleosalinity proxy for deeply buried Paleozoic and Ediacaran fossils. *Palaeogeography Palaeoclimatology Palaeoecology* 540, 109536.

Retallack, G.J., and Broz, A., 2020a, Arumberia and other Ediacaran fossils from central Australia. *Historical Biology* 32, 1755281.

Retallack G.J., and Broz, A., 2020b, Ediacaran and Cambrian paleosols in central Australia. *Palaeogeography Palaeoclimatology Palaeoecology* 560, 110047

Retallack, G.J., Matthews, N., Master, S., Khangar, R. and Khan, M., 2021, Dickinsonia discovered in India and late Ediacaran biogeography. *Gondwana Research* 90, 165–170.

Shahkarami, S., Buatois, L.A., Mángano, M.G., Hagadorn, J.W. and Almond, J., 2020. The Ediacaran–Cambrian boundary: evaluating stratigraphic completeness and the Great Unconformity. *Precambrian Research*, 345, p.105721.

Minor points

I.100 “old elephant skin” often has healed desiccation cracks, so is not merely an undermat.

We do not state that elephant skin texture is limited to being undermat, but note, with citation, that it is known from under-mat surfaces. Also, desiccation cracks can penetrate undermat layers, so there is no contradiction.

Referee: 2

Comments to the Author(s)

I have enjoyed reading about these intriguing structures. The dessication features in the Gibbett Fill Formation look convincing.

We thank the reviewer for this positive comment.

The structures from Ferryland Hill are more difficult. In particular rather hard to understand preservation in positive epirelief in argillaceous material but as mentioned in the manuscript similar preservation has been previously described. But could a cartoon be provided how the authors envisage steps in this preservation?

This is a helpful and important point. We have revised the manuscript on lines 328-333 to clarify the following points: (1) The Ferryland Head structures do show very slight positive epirelief as shown in Fig 4A and B, but generally protrude less than 1 mm from the bedding plane (probably slightly more where phosphatised); (2) This is likely because early authigenic mineralization has apparently made the streamers slightly more resistant to weathering than the surrounding fine sand; they are not mouldic fossils; (3) Where they are thick enough, streamers also project slightly downwards, consistent with early mineralization followed by compaction of the surrounding sediment, as shown by existing Figure 4B. Overall this style of preservation is quite similar to many fossil plant leaves, as detailed by Locatelli et al. (2017, the “biofilm-clay template model”), who already provide a good cartoon. We have therefore added this point and a citation to this paper (lines 335-337).

Apart from that my two main comments are on nomenclature related to Arumberia where I believe the authors need to argue this (in supplementary information). The second is on some rather washed-out images.

TAXONOMY AND NOMENCLATURE

I would suggest a second thought on the use of “pseudo-taxon” (line 147), which I quite frankly had not seen before. In the meaning of “pseudo” as something false or spurious do the authors mean that it is not a true or valid taxon?

We have removed this term as it was unnecessary and the reviewer is right to query it.

Arumberia as described by Glaessner & Walter (1975) certainly full-fills the formal criteria for the erection of a new taxon. Another question is if Arumberia is a fossil, a dubiofossil or a pseudofossil. A common convention for a taxon later found to be a pseudofossil is to maintain the name but not in italics. In the manuscript Arumberia is used without italics, which would be consistent with this, but if I read the manuscript correctly, the authors do see Arumberia as a fossil, not a pseudofossil (cf. line 66). A minority view (e.g. Stimpson et al. 2017: An ichnotaxonomic approach to wrinkled microbially induced sedimentary structures. *Ichnos* 24, 291-316, and further papers by lead author) is to consider taxa erected on MISS as ichnotaxa and so use in italics.

We thank the reviewer for focusing our attention on this issue. Our approach here was initially based on McIlroy et al. (2005), where ‘Arumberia’ is used thus, i.e., with inverted commas in Roman type. Those authors interpreted ‘Arumberia’ as a pseudofossil. We think ‘Arumberia’ is a fossil (not a pseudofossil and strictly speaking not a microbially induced sedimentary structure either) but does not necessarily correspond to any single biological taxon whereas McIlroy et al. (2005) considered it a pseudofossil and (to quote them) so “the name is retained but not as a Linnean term”. We interpret ‘Arumberia’ as a morphotype produced by a microbial mat that may involve many microbial taxa; it does not correspond either to a trace fossil (in the conventional sense) or a palaeontological taxon (in the conventional sense). One analogy might be with stromatolite form taxa, e.g., *Conophyton*, which are usually presented thus, i.e. in italics with capital initials, although the standing of these as Linnean names is a somewhat controversial question. In any case, our recognition that ‘Arumberia’ is part of a spectrum with streamers may suggest that it is not actually distinct enough to fulfil the criteria for designation as a form taxon. We have retained the Roman type but now dropped the inverted commas (in agreement with the reviewer’s suggestion below) since we do not wish to create the impression that we think Arumberia is a pseudofossil. This may not be perfect but it is probably the least controversial option and at least has the precedent of Gehling (1999).

In addition to the revised styling, we now raise this discussion point in a new section at the end of the revised Supplementary Information.

The authors state (line 193) “We consider these megascopic ribbons to be probable fossils...”, and later (lines 256-58) “We agree with previous authors that ‘Arumberia’

represents an organic matground morphotype arising from the interaction of dense biotic communities, siliciclastic sediment, and flowing water.”. There seems to be some ambiguity here if the authors really see the streamers as related with Arumberia.

We believe that the relationship is made clear through the concept of a “streamer-arumberiomorph spectrum”, illustrated in Figure 6. Possibly the term “matground” caused some confusion as it might imply something to do with the sediment underneath the mat; we have simply replaced it by “mat”.

Better use lower case “A” for “arumberiamorphs” (line 154, and throughout manuscript).

We agree — this was a mistake. We have fixed this on this line and elsewhere.

My suggestion from the above is that the authors present, probably best in the supplementary information, explanation on how they are using Arumberia. This would also make redundant the use of quotations marks.

We have followed this suggestion with gratitude.

FIGURES

Parts B, C and D in Figure 2 are rather washed-out and hard to make out detail. I assume these were images taken on a cloudy and murky day and that the authors do not have better contrast images, but could some of that be compensated for in the course of image processing?

The features themselves are unfortunately quite faint even in the best viewing conditions. We have further enhanced the contrast of panels B, C and D, and slightly shifted the field of view of D, to improve the visibility of the key features. This makes quite a big difference, especially to part B.

Part A, Figure 2. At the scale given hard to see anything of these gypsum pseudomorphs. If the idea is to show any of the morphology rather than a surface with dots then a close-up should be used.

We agree. We have “zoomed in” on this figure to show the pseudomorphs more clearly.

The above comment applies also to washed-out appearance of structures in Figure 3.

Some of these features are especially hard to see as they have very little relief, but we have further enhanced the contrast of Fig 3 panels E and G. The accompanying sketches are intended to compensate somewhat for the faintness of the features.

There is no explanation in Figure 2 caption as to significance of arrows.

We have gratefully corrected this caption (panels C and G).

OTHER

Line 16, "... these organisms ..." do not follow from preceding text.

Agreed and corrected.

Line 65, "... but there is no doubt these rocks are Ediacaran ...". I have no reason to doubt this but the sentence seems to call out for a because, which could be addressed in the supplementary information.

Indeed, this was poorly worded since the issue is not at all controversial; the import of the previous sentence (which we have clarified) was that the rocks may be older than they appear, not younger. We have revised the text here to provide supporting citations for the age of these rocks and some stratigraphic supporting evidence.

Line 166, for clarity specify already here if on bed top or bed sole, information first provided on line 175. Also add this information to the caption to figure 4.

Agreed. The line (now 258) has been amended from "bedding planes" to "bed tops" and the information has been added to figure 4.

Line 168, should "most unlike" be "unlike most"?

No, so we have reworded this to "least similar to", which seems clearer.

As I understand it all figured material is still in the field. Storage information on the thin-sections is, I believe, missing

We thank the reviewer for highlighting this. The figured material is all still in the field, except, indeed, for the thin sections. These have now been accessioned (along with the hand samples from which they were made) into the Cockburn Geological Museum at the University of Edinburgh, an appropriate collection from which they can be borrowed on request. We have added a "Sample Availability" (Lines 510-513) section giving the accession number and sample codes.

Referee: 3

Late Ediacaran life on land: desiccated microbial mats and fluvial biofilm streamers

McMahon et al.

McMahon et al. describe “macroscopic biofilm streamers” from the Ediacaran Ferryland Head Formation in Newfoundland, Canada. In addition to these really intriguing features, they also figure beautiful desiccated microbial mats and another occurrence of *Arumberia*, which is becoming an increasingly well-known surface texture from Late Neoproterozoic to lower Cambrian red beds. The descriptions of the morphology of the streamers and other surface textures are detailed and seemingly robust and will be of wide interest to readers of RSPB. Particularly important is that these structures were observed on emergent bedding planes, thus the observations further our understanding of known microbial ecosystems which had colonized the continents at this point in Earth history. I am not an expert on the occurrence of biofilm streamers, so have restricted my comments largely to the descriptive sedimentology and the inferred environments in which the streamers and other distinctive surface textures occupied.

We thank the reviewer for these positive remarks and for the rest of this helpful and well informed review, which has prompted substantial revisions to the sedimentology aspects of this paper.

I would like to see the manuscript published in RSPB, but first I feel the sedimentological framework under which these significant observations were made needs substantial improvement. Whilst I agree that the studied bedding plane of interest formed subaerially (probably as a tidally-influenced alluvial plain), I think the evidence presented for it accumulating under freshwater conditions is underwhelming. This is not to say the finding is not significant – it is – added complexity to the range of microbial ecosystems that we already know existed in continental environments at this period in Earth history. Only to say that I think there is robust sedimentological evidence suggestive of the key bedding planes accumulating in a coastal/tidally-influenced (periodically emergent) setting unlikely to be dominated by freshwater. There is actually very little sedimentological detail presented of any sort (a c. 9 m graphic log), with the reader asked to rely on a previous unpublished master’s thesis (Sala Toledo 2004) and geological notes accompanying a regional map (Williams and King, 1979) (Line 126) to confirm that their interpretation of a freshwater microbial ecosystem is robust. This is an opportunity to detail the sedimentology of an important geological succession to a wider audience and the authors must utilize it. Clearly length of the article is an issue here, but descriptions and interpretations of each sedimentary facies with accompanying figures should be added at least to the supplementary materials. The level of descriptive detail provided for the streamer morphology is superb, but this must be matched by a similar level of detail concerning the host sedimentary rocks.

The reviewer’s sedimentological criticisms and observations are very well taken and have motivated significant revisions to our manuscript, detailed in response to the points below. We do not have the resources to go back into the field to extend the sedimentary logging (particularly given restrictions due to Covid-19), but we have been able to revisit the empirical basis for our facies analysis “remotely” using field photographs (some of them kindly provided by the reviewer), prompted by the reviewer’s interpretations. Although we do not agree with the reviewer on every particular, we do agree with their major point that the sedimentology is at least as consistent with a tidal setting as it is with a fluvial one. We also agree that the palaeobiological result is exciting in either case, which implies that it is not necessary for us here to give a highly detailed account of the position of these units with respect to the fluvial-tidal transition zone — this would be of minimal interest to most readers. However, we have amended the sedimentology in the manuscript in order to present our palaeobiological findings in a sedimentological framework that is less spuriously precise than it was previously. Thus, the palaeoenvironmental interpretation is broader but more robustly supported by the evidence actually provided. More detail on these revisions is provided below.

I have split my comments into 2 “major points” (“The earliest record of an important mode of life for freshwater microbial ecosystems” and “Order of results”) as well as a series of line-by-line comments (which range from minor to quite significant). I hope this work will assist the authors in their aim of publishing details of these great bedding planes in RSPB, but I hope they also realise that first a significant amount of sedimentological detail must be added, and perhaps some revisions to the present interpretation. I emphasize that the finding is still very important and worthy of publication in RSPB even if not an unequivocal fluvial microbial ecosystem.

“The earliest record of an important mode of life for freshwater microbial ecosystems”. I find two things very concerning with this key statement: 1) evidence for a **freshwater** depositional environment; and 2) the claimed significance of the find.

1) *Evidence for freshwater*. The amount of original sedimentology presented from the succession to demonstrate the interpreted depositional environment is a little disappointing. Only a c. 9 m stratigraphic log is presented, with no accompanying photographs and no dedicated section (in the supplementary or main text) devoted to the description of each facies and interpretation. In my opinion this is insufficient in order to prove the proposed freshwater depositional environment. A recent publication in RSPB detailed non-marine (brackish-water) trilobites (<https://doi.org/10.1098/rspb.2020.2263>) in which the study included multiple

sedimentary logs in their supplementary information, as well as 2 panels of photographs illustrating key facies and a very detailed table of facies. I would expect a similar level of sedimentological detail more than achievable from the Ferryland Head outcrops. At present, this study contains one small sedimentary log and a sedimentology section spanning 9 lines (Line 116-124). There is certainly opportunity to expand this by making use of the supplementary material.

We agree with the reviewer that our sedimentological observations were not sufficient to establish a freshwater depositional environment. We have amended the relevant section of the main manuscript (which runs from line 149 to line 254) to be more inclusive of other scenarios as suggested by the reviewer, and removed references to freshwater environments throughout the paper. The log is intended to show the immediate stratigraphic context of the streamer surfaces described in the text and not to underpin a detailed palaeoenvironmental reconstruction. Further fieldwork to expand this log is not currently feasible, but in any case our revised sedimentology is less specific about the palaeoenvironmental interpretation and is well supported by observations and photographs (including new supplementary figure S4).

I do not think the buff-coloured sandstones are arkosic but arenitic.

According to the Dott classification, arkosic sandstones have greater than 25% feldspars and a higher proportion of feldspars than lithic fragments, and this tallies with our thin section and SEM observations (not shown since tangential to this paper). Even if we were to use the Folk classification the facies would be recognised as an arkose.

This is a small detail though, more important is that these sandstones almost always contain flaser cross-stratification (or at least mud-draped ripples) – see attached photograph.

We do not agree with the “almost always” characterization here; these features are quite localised.

The redbrown siltstone rich facies encase the channelized buff-coloured sandstone demonstrating that these two facies were not only coevally occurring but represent laterally adjacent settings. The buff-coloured sandstones are evidently tidal (mud-draped laminae, also bidirectional paleocurrents, sandstone bodies show evident split into mutually-evasive ebb and flood channels), so in my opinion a freshwater origin for the red-brown siltstones is extremely unlikely.

We have not seen the evidence that the reviewer interprets as recording bidirectional palaeocurrents or mutually evasive ebb and flood channels, but we have relaxed our facies interpretations to include the possibility of a tidal

influence on the environment and, throughout the manuscript, removed references to an (exclusively) fluvial setting.

Note also that (depending on channel-bed slope and tidal range), fluvial-tidal transitions may extend tens of hundreds of kilometres inland from river mouths – really difficult to prove that the red-brown siltstone facies was freshwater even if treated separately from the buff-coloured sandstones. The authors state on Line 120 that their c. 10 m (actually closer to 9 m) section is dominated by the red-brown siltstone rich facies: I can pass a couple more comments on the descriptive sedimentology and possible interpretation of this facies which I hope are useful for the authors when making their revision. Note importantly though that it is the association between the redbrown siltstones and the clearly tidal buff-coloured sandstone facies is key to its interpretation (another reason why the single 9 m thick log would ideally be expanded).

We respond further to the reviewer's comments on the sedimentology below.

The red-coloured, fine to very fine-grained, predominantly parallel-laminated silty sandstones display current ripple cross-lamination and soft sediment deformation.

The presence of these features is already noted within our description.

They occur in association with channelized (“buff-coloured”) sandstones predominantly filled with mud-draped ripple cross-laminae. Also intercalated are redcoloured mudstones which host *Arumberia* and the described “streamers”. Intercalated sandstones show current- and wave-formed ripple lamination. Desiccated polygons are also widespread, in addition to other surface textures showing evidence of emergence (e.g., adhesion marks, gypsum pseudomorphs). I suspect the red-brown siltstone rich facies preserves a tidally-influenced alluvial plain. Periodic emergence is supported by the widespread presence of desiccation cracks, whereas other surface textures indicate periodic submergence (e.g., waveformed ripple lamination). Whereas the buff-coloured sandstones likely formed in tidally-influenced fluvial channels/tidal channels/estuarine channels, the red silty sandstones represent unconfined deposition. Water escape structures suggest periodic saturation (fluctuating water table?). The red mudstones which preserve *Arumberia*/streamers represent deposition on a tidally-influenced alluvial plain (or a playa/mudflat?). The distinction between these would be that alluvial plains are fed by river systems (perhaps the buff-coloured sandstone facies?). Association with tidal sedimentary structures goes against a freshwater origin for the streamer-hosting bed. Whilst the beds are definitely continental/non-marine – stating that they are freshwater, and therefore that the observed streamers preserve a freshwater microbial ecosystem, is a step too far (and I don't think valid). This becomes very significant in the discussion, where the term “fluvial” is used repeatedly (and I think incorrectly). The

authors are free to disagree with my opinion on the unit's sedimentology, but if they want to press their case for a freshwater origin and "fluvial streamers" then substantially more sedimentological detail is required. Personally, I think the term "freshwater" should simply be disregarded in all instances – and replaced with "continental" or "terrestrial". This would still be a significant find.

This account is very largely — but not completely — in accordance with our own observations and interpretations; but where we differ it is because we have not seen some of these features or consider them more open to interpretation (e.g., we would not describe the buff sandstones as "predominantly filled" with mud-draped ripple cross-laminae). Thus we have relaxed our palaeoenvironmental interpretation to accommodate the possibility that the reviewer is correct about a tidal depositional environment. We are grateful to the reviewer for outlining this interpretation, and for noting that it does not invalidate our description of the setting as terrestrial or non-marine. As such we have amended the description and conclusion to remove mention of fluvial or river environments that may imply a definite freshwater depositional environment. We take the view that diagnosing a specifically tidal facies would be overinterpretative. Rather, as stated, we prefer the Reviewer's previous constructive suggestion of recognising the uncertainty in the environmental interpretation, insofar as there is a distinct possibility of tidal influence on these deposits. At the recommendation of the reviewer we have therefore:

- 1. We have replaced the term "freshwater" with "terrestrial" (throughout).**
- 2. Amended mention of 'braided river system' to 'braided channel system' (on line 171).**
- 3. Added the association of channels to the buff sandstone facies, and removed the interpretation of crevasse-splay deposits with this unit (on line 172).**
- 4. Regarding the red-brown muddy facies, we have removed mention of them being 'overlying' (when compared to the sandstone facies), and noted in the interpretation that they represent environments of 'unconfined deposition' (on lines 173-174).**
- 5. Added an additional sentence noting that "further work is required to understand the influence of tidal processes on the section, and therefore the salinity of the palaeohabitat occupied by the fossils reported here" (lines 189-191).**

As noted by the reviewer, our new less specific conclusion of a non-marine environment that may have been tidally-influenced does not detract from the significance of the palaeontological find.

As a minimum: description and interpretation for each facies in the Ferryland Head Formation is required – not simply asking the reader to refer to two previous reports (Line 126), one a report accompanying a geological map and the other a master’s thesis – both very hard to track down! The extent of the facies description of the Ferryland Head Formation in the former of these is 6 lines – and hardly providing a compelling case of deposition “in fluvial conditions” (Line 127). The original authors in that study do not describe the formation as representing fluvial conditions but forming beds “that appear to be channel sand deposits” (so why not tidal/estuarine/tidally-influenced etc. Not necessarily fluvial). Ideally a log would also be presented throughout the entire exposed section – this doesn’t necessarily need to be as high resolution as the 9 m stratigraphic log presented, which is the key section undoubtedly, but it should be there. Perhaps there is one already available in the Matthews 2011 thesis?

We regret that the ongoing pandemic precludes us from participating in further fieldwork to provide the additional information requested by the reviewer. However, noting the changes we have made based on the welcome suggestions to be less specific in our environmental interpretations, our revised conclusions are now suitably supported by the evidence provided.

2) *Significance of find*: The paper frames the find as a significant “earliest record of an important mode of life for freshwater microbial ecosystems”. There is no clarification on what this “mode of life” actually is? Do the authors specifically mean “streamers”: as in, this is the earliest “fluvial biofilm streamers” reported? Or do they mean any form of freshwater microbial life. Notwithstanding my above concerns of the studied bedding plane hosting the streamers as having a freshwater origin, if the latter is intended then there are a number of outstanding citations of earlier reports. For example, microbially-induced sedimentary structures from the more unequivocally freshwater Mesoproterozoic Meall Dearg Formation in Scotland (Prave, 2002, [https://doi.org/10.1130/0091-7613\(2002\)030%3C0811:LOLITP%3E2.0.CO;2](https://doi.org/10.1130/0091-7613(2002)030%3C0811:LOLITP%3E2.0.CO;2) - see also McMahon and Davies, 2018, <https://doi.org/10.1144/jgs2017-012> - though this second study is far less important). Also, fluviolacustrine stromatolites from the Mesoproterozoic Copper Harbour Formation (Fedorchuk et al. 2016 - <https://doi.org/10.1016/j.precamres.2016.01.015>).

There are three additional MISS reports from earlier strata reported in the cited Davies et al. (2016, ESR) work, their Table 1 – though I cannot comment on the veracity of any of these claims.

The mode of life we intended to highlight was the formation of large streamers in a current rather than just MISS. We have now clarified the wording as follows: “...best interpreted as fossilised current-induced biofilm streamers, the earliest

record of an important mode of life (macroscopic streamer formation) for terrestrial microbial ecosystems”. (Line 38).

Order of results

I am curious as to why the authors have decided to present the desiccated microbial mats with “flipovers” first, followed by *Arumberia*, followed by the macroscopic biofilm streamers. It seems to me that the last of these is by far the most significant finding – if indeed the authors are able to prove that these biofilms did occupy freshwater fluvial environments (and even if not, I think it is still the most noteworthy find). There have been a number of MISS reports from tidal environments earlier than this study (see Table 1 in the cited ESR paper by Davies et al.) and no less than 39 *Arumberia* case studies in total. What’s truly novel about this study is the report of the streamers. That’s not to say that the *Arumberia* and MISS reports aren’t interesting – they are – but I would have thought secondary. Of course, if the authors can demonstrate why they chose to present the results in the order they did (stratigraphic?) of course that is OK, just seemed peculiar whilst reading.

We agree with the reviewer that the streamers are our most significant finding and (gratefully) also agree that they are significant regardless of their position in the fluvial-tidal transition zone. However, the flipovers we report belong to the same story — microbial palaeobiology of emergent/terrestrial settings in the Ediacaran (of Newfoundland) — which indeed we are telling in stratigraphic sequence. The flipovers may not be as novel as the streamers, but they are probably the best examples that have been found in the fossil record (the quality of mouldic preservation seems almost unimprovable) and deserve to be well known. To integrate them better into the paper we have added mention of them to the abstract.

Line-by-line:

Line 17 (and elsewhere). “Fluvially-influenced”: What do the authors mean by this term? To my understanding it has no grounding in the literature until this point. If used it would have a significantly wide reach surely? Floodplains are fluvially-influenced, as well as the main channel and all secondary conduits, crevasse splays etc. Are oceans and lakes also “fluvially-influenced” if the rivers feed them? Perhaps the authors mean that tidal/marine processes are the dominant depositional process, with only a slight involvement of the feeding fluvial system influencing the final preservation? I suspect this is not the case though if the authors do believe the studied bedding plane formed in freshwater.

We have removed this term.

Line 40. Not intending to push my own work, but McMahon et al. (<https://doi.org/10.2110/jsr.2020.029>) contains substantially more details on the marine fossiliferous environments in comparison to the studies presently listed. Matthews et al. (2020) is a great study, but my understanding is that it provides information on chronostratigraphy, not a detailed sedimentological account of the prevailing depositional environments. Could also consider citing good papers on both the White Sea Biota in Russia (e.g., Grazhdankin, D.V., 2003. Structure and depositional environment of the Vendian Complex in the southeastern White Sea area. *Stratigraphy and Geological Correlation*, 11(4), pp.313-331) and Nama Biota in Namibia (e.g., Maloney, K.M., Boag, T.H., Facciol, A.J., Gibson, B.M., Cribb, A., Koester, B.E., Kenchington, C.G., Racicot, R.A., Darroch, S.A. and Laflamme, M., 2020. Paleoenvironmental analysis of Ernie-tta-bearing Ediacaran deposits in southern Namibia. *Palaeogeography, Palaeoclimatology, Palaeoecology*, 556, p.109884).

We agree, and have added all of these useful references (and removed Matthews et al 2020 from this line).

Line 42. Is this true? Davies et al. 2016 go to great lengths to highlight MISS distribution throughout the Precambrian and Phanerozoic (Section 2) and discuss how any increased MISS in the Ediacaran and Cambrian may be in part reflect a sampling bias due to greater interest in surface textures from this time period in comparison to, for example, the Silurian. It would be appropriate to change the sentence to “are, however, **commonly reported** in Late Ediacaran”. Note also that Table 1 contains the fluvial MISS reported by Prave (2002) from the Mesoproterozoic in addition to three other earlier MISS reports (all Mesoproterozoic).

We agree and have corrected this oversight.

Kolesnikov (et al. 2017) is also a very curious reference to add in at this point. The paper discusses arumberia-like surface textures found on a modern saltern and make some comparisons between those and *Arumberia* in two Russian formations – neither of which have been described as occurring in “fluvially-influenced” depositional environments.

The citation to Kolesnikov was in support of the “tidal” part of this sentence so we retain it.

Line 53. Is there a citation that could be used for the statement here about the depositional environments of the Caphhayden and Gibbett Hill formations?

Thank you for raising this. We have added a citation.

Line 57. I would consider substituting “fluvially-influenced” to “records the onset of periodically emergent conditions”. This is far more accurate. Note also that marine red

beds can also occur; for example, the Caerfai Bay Shales in Wales.

Done. Point taken, but the red beds at Ferryland Head are not marine.

Line 81/Figure 1A. Beautiful!

Thanks!

Line 101. If it's well known could we perhaps have a couple of more references?

We have added two more references to papers referring to elephant skin texture on under-mat surfaces.

Line 126. An unpublished thesis and a paragraph in a mapping report can hardly be considered a robust summary, given the significance of the claimed discovery. Additional sedimentological details should be presented here – or at the very least in the supplementary materials. See major comments.

This was a fair point. Having softened the interpretation from “fluvial” to “non-marine” and expanded on and refined the sedimentology as detailed in our responses to other points, the palaeoenvironmental interpretation is now sufficiently broad as to be justified by the evidence offered in the present paper.

Line 127. “We consider the laterally extensive bed geometries and observed sedimentary structures to be indicative of deposition within a braided river system”. Meandering channels can have extensive bed geometries as well? As for the observed sedimentary structures – very few have been presented?

As noted, this conclusion is no longer made as we have broadened our environmental interpretation.

Line 129. “The buff sandstone facies is here considered to be associated with proximal over-bank and crevasse-splay deposits, and transitions into the finer- grained facies”. I don't agree with this. The buffcoloured sandstones form low-relief channels, contain mud-draped ripple cross laminae, as well as subordinate dune cross-strata. These are in my mind shallow and wide tidally-influenced channels. In order to show deposition occurring as crevasse-splays considerably more detail is required – a figure showing key faces, a photograph showing the extensive sand-body architecture etc.

<https://doi.org/10.1016/j.sedgeo.2017.02.003> - This is a very good paper on crevasse-splay deposition that lays out robust criteria on recognizing such styles of deposition in ancient stratigraphy.

The text in this section has been edited as noted in response to the reviewer's major comment 1. We have broadened the interpretation to include channelised systems, while being careful not to infer a purely fluvial interpretation. Since we no longer argue specifically for crevasse-splay

deposition we trust that the extra detail requested is not necessary. Nevertheless, we have added a supplementary photograph (new Figure S3) that provides an overview of the studied units.

Line 131. The red-brown facies encase the buff-coloured sandstones – and are not just “overlying” it.

We have removed the word ‘overlying’.

Line 132. Which channels? This interpretation of the facies relies on only the overbanks being preserved throughout the entire exposed section? Did the authors observe channels anywhere in the c. 250 metre thick Ferryland Head Formation? I would suggest that the red-brown facies represents the “overbanks/floodplains/alluvial plains”, with the buff-coloured sandstones representing the channels?

Our revisions to this section make it clear that, in agreement with the reviewer, some of the buff sandstones may represent channels, and that these may not exclusively be fluvial channels. The red-brown facies is now described as representing unconfined deposition, in agreement with the reviewer. We hope that the revised text is now acceptable to the reviewer.

Line 132. “periodic wetting and drying cycles”. What drives these cycles? Again – the observation fits well within a tidal setting. Wetting and drying driven as current drains water away following an interval of elevated water level (e.g., in the receding waters of a tidal prism).

We think the broader and more inclusive environmental interpretation we now provide allows for this. Wetting and drying can occur in tidal settings, but also occurs elsewhere.

I’m also confused as to why the red-brown facies has been introduced here briefly, before a switch is made back to the buff-coloured sandstones, only for the red-brown facies to be introduced properly in the next paragraph?

The reviewer is correct that this was confusingly organised. We have reordered the text so that a discussion of the general stratigraphy and depositional environment precedes the focus on the siltstone-rich facies in which the streamers are found.

Line 134. I don’t follow this logic at all. 1) “Wave ripples are consistent with this facies” – What does that mean? Wave ripples occur within this facies? Whilst it is possible that the wave ripples formed in ephemeral braidplain lakes agitated by wind, there are copious other environments in which such structures could form. I can’t help but feel that this reference was cotton-picked because it provided one mechanism which fitted the envisioned freshwater origin.

The reviewer raises an important point with regard to our unhelpful wording. This has been amended to clarify that wave ripples are found within this facies. We have added the word 'potentially' to note that the braidplain conclusion is but one of the environments in which this structure would be found.

Lien 135-137. Current ripples can form in a whole host of depositional environments. How do mud-draped current ripples (see attached photograph) develop in an unconfined crevasse splay?

We have removed the mention of crevasse splay deposits as it was overinterpretative. (Current ripples are common in crevasse splays and mud drapes would simply reflect waning currents.)

Line 138. I don't think it is necessary to make a link to the floodplain fines element of a geological textbook here.

Noting the link to previous comments as well, we have removed this unnecessary sentence at the suggestion of the reviewer.

Line 139-141. This seems random? Why does the reader need to know this statement about meandering river facies?

Noting our now broader and not exclusively fluvial interpretation we agree with the reviewer that this sentence is no longer necessary.

Line 141. Please provide some photographs of the key characteristics of the buff-coloured sandstone facies.

This paper is not primarily a sedimentological study and our revised, less specific interpretation of the depositional environment is supported by the evidence and descriptions already provided. Nevertheless, we have added a supplementary photograph (new Figure S1) that provides an overview of the studied units.

Line 149. I don't think *Arumberia* has been definitively proven to occur in fluvial facies

We have amended 'fluvial' to 'alluvial plain' as is mentioned in the reviewer's 2017 paper.

Line 157. Could the authors pass comment on whether or not they think splitting *Arumberia* down to the species level is appropriate? Or whether they think different forms may reflect different burial conditions/moments of interment? Figure 6 does a

really nice job, and suggests to me that the authors believe it is purely to do with the prevailing current/sedimentological conditions.

To be clear, Figure 6 presents a simple model for consideration and may very well not tell the whole story. We suspect that splitting *Arumberia* to “species level” with latin binomials is likely to cause confusion because the various forms probably fall on a spectrum (this is a prediction of the hypothesis that these forms are not simply different organisms) — and have added a new note on nomenclature to the supplement.

Figure 3. Really great bedding plane and nicely presented figure.

Thank you for this comment.

Line 195. Earlier the biogenic *Arumberia* was described as being equidistant. Don't think anyone would believe these are ripples – think sentence can be omitted. Though I appreciate the caution.

That earlier reference was to the classic rugose form of *Arumberia* and not to these streamers. In any case, we have followed the suggestion and omitted the sentence.

Line 232. Are these streamers the same as the structures described as *Arumberia Beckeri* in Kolesnikov et al., 2012? (their figure 3).

A bit different in scale and morphology but probably part of the same spectrum, certainly preserved in the same way as the material at Ferryland Head. We now address *beckeri* briefly at the end of the Results section, where the revised text reads: *The mm–cm-wide longitudinally striated ribbons of ‘Arumberia ollii’, and the parallel or fanning, 0.5 mm-diameter curved filaments of ‘Arumberia beckeri’, both found as aluminosilicified compressions from the Ediacaran of the Central and South Urals (Kolesnikov et al., 2012) can also be interpreted as small streamers similar to those at Ferryland Head. Kolesnikov et al. (2012) suggested that these forms might be taphonomic variants of each other, and Kolesnikov et al. (2017) proposed that they were both unrelated to ‘Arumberia’ since they consist of discrete elements rather than textures on a mat. Our model suggests that they probably belong on a spectrum with arumberiomorphs, and predicts that intermediates may exist.*

Line 283 and 289. “Alluvial” probably more appropriate than “Fluvial” in both instances. I don't think compelling evidence can be provided to demonstrate that the succession is fluvial in origin.

The term “fluvial” has been removed.

Line 304. Also *Arumberia Beckeri*? See above.

As above.

Line 309. Alluvial, not fluvial.

The term “fluvial” has been removed.

Line 309. I think a sentence is needed somewhere to demonstrate that these were not wholly non-marine environments, but periodically emergent environments. Life was clearly able to take hold and survive subaerially – that’s the significance.

As outlined elsewhere we have amended our text to broaden our environmental interpretation, though we don’t go as far as the reviewer is here in completely ruling out a non-marine environment.

Line 312. Do coastal examples also exist?

Coastal examples do exist and can easily be found in the field although they are surprisingly rarely mentioned in the literature. One published example is described as “tufts” and “floating cyanobacterial filaments” figured by Lakhdar et al. (new citation) but shows clear parallel alignment (their Fig 9C) from tidal flats on the Mediterranean coastline of Tunisia. These represent small streamers. This example has now been cited here.

Line 321. Fluvioterrestrial – Why introduce this term at this point in the manuscript?

This term has been removed here and again on the last line.

Line 327. I really think this should read some “subaerial” settings, or some “emergent” settings. I don’t think a fluvial interpretation will stand the test of time.

This term has been removed.

Line 329. Fluvio-tidal – another new term, but probably the most accurate used so far to describe the Ferryland Head succession.

We have retained this term since it is appropriately broad.

Line 331-333. Is it necessary to note this? Note, McMahon et al. 2017 (already cited) state that the fluvial rock record will always average to the maximum normal deposition, and studies of prevegetation fluvial stratigraphy by and large have shown that this

threshold exceeds the erosive threshold of surficial biofilms. Any eco-engineering capacity they provide daily is not archived in the ancient sedimentary rock record (it is lost when the biofilms ultimately fail). That said, present work in press in another journal have detailed how in pre-vegetation, low-gradient, muddy, coastal systems, engineering effects caused by microbial communities can have a greater chance of preservation. The cited study by Ielpi et al. 2018 only speculate on the impact of microbial life on alluvial architecture (there Section 5.4 “outstanding questions”). Another study perhaps of use is this recent work in ESPL, which concerns the role of microbiota in a modern coastal system:

<https://doi.org/10.25850/nioz/7b.b.m.>

We think this is a worthwhile thing to note, that adds to the impact of the paper by highlighting the relevance of microbial communities for understanding pre-vegetation sedimentology. With thanks, we have added the new citation as another example of microbial influence on sedimentary architecture in a tidal system.

I have not checked the reference list.

Finally, I believe the conceptual model of the relationship between the streamers and more conventional *Arumberia* is superb and very valuable to the scientific community. I emphasize that I hope my criticisms are taken as constructive. With a sounder sedimentological framework, I believe the study will make a strong contribution to RSPB.

We thank the reviewer for these kind remarks, for supporting publication in RSPB, and for their constructive and detailed review, which has particularly improved our sedimentological interpretations.

The authors are welcome to contact me if they have any questions relating to this review.

Thanks again to the reviewer. We are of course happy to consider further revisions if these do not go far enough.

William McMahon

Appendix C

McMahon et al.

Late Ediacaran life on land: desiccated microbial mats and large biofilm streamers

I thank the authors for making substantial and good revisions to their text following the previous round of review. I believe the product is more robust and, following a series of minor corrections, should be publishable in Proceedings of the Royal Society B. An improved documentation of the early steps of the terrestrialization of the continents is greatly beneficial for a wide number of studies across the Earth Sciences. New ideas on the development and preservation of *Arumberia* are also good to add into the current mix.

All my minor comments are listed line-by-line below. In addition to these, I would once again suggest that the authors consider leading the manuscript with their description of the biofilm streamers, with the desiccated mats in the Gibbett Hill Formation coming later. Whilst I appreciate that the authors chose to present their results in stratigraphic order, it just comes across as a prelude to the main act. Furthermore, no descriptive sedimentology of the Gibbett Hill Formation is presented anywhere in the manuscript, in comparison to the Ferryland Head Formation, where the sedimentology is introduced in some detail. I think overall the contribution will be more concise if reorganizing to lead with the detailed descriptions and interpretations of the Ferryland Head streamers, with the Gibbett Hill mats coming afterwards as a very worthwhile add on. Of course if the authors feel very strongly that the desiccated mats should come first, it's not my place to say otherwise.

Line by Line comments

Lines 13 and 25. In keeping with the changes made elsewhere in the manuscript, "Non-marine" should really become "Continental" or "Terrestrial" (i.e., occurring on land). Non-marine suggests no marine influence, whereas I'd argue sedimentological evidence attests there was. Terrestrial ecosystem is used on line 24, so suggest the authors stick with this.

Line 18 (and 80). Circling back on discussions of fluvial and tidal influence elsewhere, how can it be known that the mouldically preserved desiccated mats were tidal? They probably are, but no sedimentary facies work from the Gibbett Hill Formation is presented throughout the manuscript. I would argue that making a visual comparison to a modern analogous environment (line 99) is insufficient. We know the bedding plane is emergent from visual comparison, but inferring a tidal environment requires another step.

Line 38. If removing "non-marine" from the abstract, could consider replacing "terrestrial" with this term here in order to really make this point.

Line 49. I think it is sensible to add “their Table 1” after the Davies et al reference. One of the key take home messages in that paper is that MISS are pretty evenly distributed throughout the Ediacaran and Phanerozoic. At present this sentence could mistakenly be misconstrued that Davies et al. hold the school of thought that there is some sort of MISS bias in the Ediacaran. That amended reference on its own would probably be enough here – not sure the relevance of modern salterns in Kolesnikov et al., 2017 here, but realise this was discussed at previous round of review (otherwise why not cite each of the listed MISS occurrences in fluvial-tidal rocks listed in the table in Davies et al.).

Line 60. As commenting on regional place names, a good place to refer to Supp. Figure 1 perhaps?

Line 75. Check convention, but I don’t believe “Group” should be capitalized here.

Line 79. Is that comma required? Perhaps stylistic preference.

Line 82. Delete “here”? The sentence already starts with this, and think reads better without the repetition.

Line 86. Does “Ediacaran” need to be in this heading? No other time periods are being discussed in this manuscript.

Line 87. “Gibbett”

Line 90. Check convention, but I don’t believe formation should be capitalized here.

Line 91. Presuming “typical” relates to quantity, perhaps change to “the more abundant” sandstones.

Line 108. Not entirely certain this “re-evaluation” statement is fair. Myrow 1995 provides an extensive overview of the regional stratigraphy. I don’t think anywhere is a “shallowing-upwards” model proposed – but to declare the work as “simple” is probably unnecessary considering the work did not attempt a detailed look at the unit’s sedimentology. Better to simply leave this as: “The Gibbett Hill Formation has always been considered permanently subaqueous (REFs), with evidence of emergence not occurring until the Ferryland Head Formation (REFs). The report of this new bedding plane prompts re-evaluation, suggesting at least one cycle of deepening and then shallowing following this deposit.” (or something along these lines). Note Myrow is not in the reference list.

Line 115. Gehling et al. 2009 (?)

Line 117. In my previous review I didn’t clock the local importance of recognizing evidence of emergence in the Gibbett Hill Formation, nice.

Line 132. Delete extra space before lamination.

Line 137 “Soft-sediment”

Line 138. Convolute-lamination (with hyphen for consistency with line 132) is a form of soft-sediment deformation. So why specify both? Were other forms of soft-sediment deformation observed? If so specify.

Line 143. As discussed, suspect tidally-influenced. Certainly envision the relevant desiccated mudstones hosting the streamers and *Arumberia* as a tidally-influenced alluvial plain. Perhaps “continental” instead of “non-marine”? If simply reporting what the previous researchers have concluded, then I would recommend an additional sentence building on their work based on your observations.

Line 144. Deducing channel planform from preserved aspect ratios is not straightforward, though I would actually suspect that being able to recognize channel margins at a scale subordinate to the “normal” sedimentary outcrop dimensions (and Bread and Cheese Cove at least) is evidence against a braided planform (where autogenic reorganization of the bars, particularly in a sandy system, would lead to the reworking of the preserved channel margins and far more laterally extensive tabular sandstone bodies). This is speculative though with the present evidence. Up to the authors, but my recommendation would be to simply remove this sentence – the bed geometries aren’t that laterally extensive in comparison to many channelized/non-channelized outcrops worldwide. Check out the classic work of Gibling (2006) below – the figures not only emphasize the enormity of other sandstone bodies, but also the extent to which aspect ratios of sandstone bodies might be studied.

- Gibling, M.R., 2006. Width and thickness of fluvial channel bodies and valley fills in the geological record: a literature compilation and classification. *Journal of sedimentary Research*, 76(5), pp.731-770.

Line 146. Overbank should be one word.

Line 147. Add “laterally” before transitions.

Line 149. More distal to the channels? So the authors believe the channelized buff-coloured sandstone units pass down-system into the red-brown *Arumberia*-bearing siltstones? I don’t believe this is the case, with them instead being laterally adjacent environments (channel and alluvial plain respectively). The red-siltstones encase the buff-coloured sandstones demonstrating this. More likely you will pass down system into permanently (or dominantly) subaqueous facies – i.e., the Gibbett Hill Formation.

Line 152. I still find this a quite bizarre choice of reference. Wave ripples are a prominent part of the “floodbasin” facies of Lebeau and Ielpi, but not actually at the sections featured in their study, which are dominantly aeolian. Authors should keep if they think appropriate, but suspect far more relevant to cite some classic sedimentological work actually discussing the formation of wave ripples in some detail: Reineck and Singh *Depositional Sedimentary Environments*, for example.

Lines 190 and 201. Why the change from “bedding planes” to “bed tops”? Think the original works better.

Line 192 – 195. Can this sentence be broken up into two? It's quite wordy at present considering it is the introduction to the key bedding plane in the study. Took me a couple of passes at it to make sense of it.

Line 212. Journal preference, but suspect "1" should become "one"

Line 220. Could replace "." With ":" and then list reasons(?).

Line 224. There are *Manchuriophycus* cracks in the Gibbett Hill Formation. The features described here are of course distinct, but if the authors wish to mention *Manchuriophycus* here and have photographs of these to hand they could consider adding it as a supplementary image.

Line 231. "synaeresis"

Line 231. Is the second part of this sentence "or indeed dewatering structures or other sedimentary structures" really necessary? Visual dissimilarity alone proves this isn't the case. Just seems a little odd to specify why not some particular types of sedimentary structures, and then follow this up by saying "or indeed any sedimentary structure".

Line 233. "demonstrate" instead of "show"(?).

Line 235. Perhaps "Weathering may have enhanced the visual contrast between the better indurated," is more concise.

Line 245. "five" (?)

Line 246. Change "above it" to "higher in the section"? Longer but reads better I believe.

Line 254. "the" underlying laminae

Line 281. Perhaps change McMahon et al., 2017 to McMahon et al., 2021. Maybe appropriate also to credit Bland, 1984 here – seeing as this interpretation largely aligns with his earlier viewpoint.

- Bland, B.H., 1984. *Arumberia* Glaessner & Walter, a review of its potential for correlation in the region of the Precambrian–Cambrian boundary. *Geological Magazine*, 121(6), pp.625-633.

Line 302. I think the sentence reads better without the "the" before preserved(?).

Line 324. Continental/Terrestrial/Periodically emergent

Line 326. If making a suggestion to someone else's work, would it not be more appropriate to reword to something like "suggested here to be probable fluvial streamers"? Unless maybe you've seen them first hand, in which case could mention as such.

Line 335. "that" can be observed

Line 344. Really nice.

Line 356. If "tidal" is lost at the earlier description of these, ensure removed here also.

Lines 383 – 384. This sentence needs changing up if it's being retained.

- Floodplains should be one word.
- Van der Visjel et al., 2019 refers to modern biofilms, as a possible analogue for Precambrian settings in some instances.
- Ielpi et al., 2018 (as discussed previously) wholly speculate on the possible impact of microbial mats on architecture.
- So as a minimum the sentence should read “We note that microbial communities can increase sediment cohesion on channel-bounding floodplains (van der Visjel et al., 2019) and thus “potentially” also affect the “preserved” architecture of fluvio-tidal systems (Ielpi et al., 2018).
- More relevant would be to cite a recent modelling paper led by Muriel Brückner (below) (in addition to or instead of the work by Ielpi et al). Numerical models are employed to study how various stages of plant evolution might affect sedimentary processes and preservation in fluvio-tidal environments. The first “vegetation stage” ran inputs with only microbiota present, with the results suggesting extensive colonization in floodplain tracts at certain points in the fluvial-marine transition zone, with resultant heightened levels of mud deposition facilitating the preservation of more muddy coastal plain facies.
- To my knowledge this is the only study to date that has done something tangible in terms of linking microbial mats to shifts in architecture.
 - Brückner, M.Z., McMahon, W.J. and Kleinans, M.G., 2021. Muddying the Waters: Modeling the Effects of Early Land Plants in Paleozoic Estuaries. *Palaios*, 36(5), pp.173-181.
- As I mentioned in my previous review, no studies of the rock record have demonstrably shown an influence of microbial mats on *preserved architecture*. Mat-influence appears to be below the average norm of deposition and thus any original sedimentological influence is ultimately lost and not preserved.

Line 402. “studied”

Line 412. “William” (thanks)

Line 432. “ripples” should be “ripple marks” (preserved not extant)

Line 455. “Bed top vs Bedding plane”(?)

There are lots of missing references. Myrow 1995, a couple of Gehling papers, Billings, 1872 and more.

A complete sweep of the manuscript is required.

The supplementary note 2, *Arumberia* nomenclature, is an excellent addition to the manuscript.

William McMahon

Appendix D

Associate Editor

With respect to aluminosilicification, as is addressed here, I would also like to highlight the recent publication by Becker Kerber, that advocate a secondary replacement of metamorphosed organic material by aluminosilicates.

I should also stress in relation to the study by Locatelli et al. (2017) that in my own experience looking at terrestrial soft tissue fossils where the rocks evidently have experienced some weathering due to flow of meteoric water you also see films replacing the organic material partly by clay minerals and partly by iron oxides.

My suggestion would be to perhaps either address why the observed microbial mats should have been preserved by primary aluminosilicification, or let it be open to having been incurred during later stage volatilisation of organically preserved microbial mat and infilling with aluminosilicates.

Sedimentological observations may allow to distinguish between these two by analysing the clay mineral fabric under secondary electron microscopy/back scattered electrons? However, it may not be of primary interest for the nature of the current study in which case reverting to simply acknowledging the potential primary and secondary mechanisms for the observed phenomenon would be a simpler solution.

Thank you for these thoughtful suggestions, which we have followed. Our revised text acknowledges plainly that we cannot constrain the timing of clay replacement (page 10). To provide a more balanced and detailed comment on the taphonomy without exceeding the page limit, we also added a new Supplementary Note incorporating some of what was previously in the main text. The new Supplementary Note acknowledges the possibility of “late-stage volatilisation of organically preserved material and/or infilling of secondary porosity by aluminosilicates” with a citation to the Becker-Kerber paper suggested here as well as citations suggested on the same theme by reviewer 4.

Other minor revisions connected with our discussion of the taphonomy are detailed below in our response to reviewer 4’s comments. Arumberiamorph taphonomy is remarkably consistent globally, a phenomenon that probably merits more attention than we can give it here. We consider this beyond the scope of the present article.

Reviewer 1

This is improved however there is one issue and one minor glitch still

The main problem is that Arumberia is not universally regarded as a mat structure. The following paper should be cited and reasons given why interpretation of Arumberia as a discrete fossil is incorrect.

Retallack, G.J., and Broz, A, 2020, Arumberia and other Ediacaran fossils from central Australia. *Historical Biology* 32, 1755281.

In response to this, we have added the following remark towards the bottom of page 13: “Interpretations of Arumberia as a discrete macroorganism (e.g., Retallack & Broz, 2020) are implausible given its indistinct edges and the gradational transitions observed between different morphologies over short distances (McMahon et al., 2021).”

Why is the title to Tarhan et al. (2017 in all caps?

Titles of Palaios papers appear in all-caps and are often cited as such but the reviewer is probably correct that they shouldn't be; we have edited this as well as other Palaios papers cited (but will defer to the RSPB copy-editors if it needs to be changed back).

Reviewer 2

No comments to address from referee 2.

Reviewer 3: comments extracted from PDF:

Late Ediacaran life on land: desiccated microbial mats and large biofilm streamers
I thank the authors for making substantial and good revisions to their text following the previous round of review. I believe the product is more robust and, following a series of minor corrections, should be publishable in *Proceedings of the Royal Society B*. An improved documentation of the early steps of the terrestrialization of the continents is greatly beneficial for a wide number of studies across the Earth Sciences. New ideas on the development and preservation of Arumberia are also good to add into the current mix.

We thank the reviewer for the considerable investment of time and care in their feedback on our paper.

All my minor comments are listed line-by-line below. In addition to these, I would once again suggest that the authors consider leading the manuscript with their description of the biofilm streamers, with the desiccated mats in the Gibbett Hill Formation coming later. Whilst I appreciate that the authors chose to present their results in stratigraphic order, it just comes across as a prelude to the main act. Furthermore, no descriptive sedimentology of the Gibbett Hill Formation is presented anywhere in the manuscript, in comparison to the Ferryland Head Formation, where the sedimentology is introduced in some detail. I think overall the contribution will be more concise if reorganizing to lead with the detailed descriptions and interpretations of the Ferryland Head streamers, with the Gibbett Hill mats coming afterwards as a very worthwhile add on. Of course if the authors feel very strongly that the desiccated mats should come first, it's not my place to say otherwise.

If the Gibbet Hill mats go before the streamers they may seem like a prelude but if they go afterwards they may seem like an afterthought. One can argue either way about this; the stratigraphy provides a good reason for keeping the order as it is. We now briefly elaborate on the sedimentology of the Gibbett Hill Formation to support the tidal interpretation of the mat without sacrificing too much concision (see Line 18 comment below).

Line by Line comments

Lines 13 and 25. In keeping with the changes made elsewhere in the manuscript, “Non-marine” should really become “Continental” or “Terrestrial” (i.e., occurring on land). Non-marine suggests no marine influence, whereas I’d argue sedimentological evidence attests there was. Terrestrial ecosystem is used on line 24, so suggest the authors stick with this.

In the previous round of review, this reviewer stated that the beds were “definitely continental/non-marine” so this is a bit of a change of tack. But to avoid any possible ambiguity, we have revised the wording as follows. The first “non-marine” queried here has now been replaced by “on land”. The second one is now “non-marine/transitional”. We retain the other instances of “non-marine” since they are not referring exclusively or specifically to Ferryland Head.

Line 18 (and 80). Circling back on discussions of fluvial and tidal influence elsewhere, how can it be known that the mouldically preserved desiccated mats were tidal? They probably are, but no sedimentary facies work from the Gibbett Hill Formation is presented throughout the manuscript. I would argue that making a visual comparison to a modern analogous environment (line 99) is insufficient. We know the bedding plane is emergent from visual comparison, but inferring a tidal environment requires another step.

The reviewer makes a solid point here that motivated some small revisions. We have added the following text at the bottom of page 4: “These host sharp-crested, symmetrical ripple marks, which we interpret as the product of wave activity as well as multiple horizons with sedimentary cracks suggestive of desiccation” supported by a new supplementary figure showing the ripple marks and desiccation cracks.

At the top of page 6, we have clarified: “noting previous shallow marine interpretations of the Gibbett Hill Formation, the presence of wave ripple marks and desiccation cracks on multiple horizons, and the new evidence for emergence provided by these dessicated microbial mats, it seems likely that the latter inhabited a tidal setting.” It is hard to see how the setting could be anything but tidal here — the mat was evidently exposed by a freak event.

Line 38. If removing “non-marine” from the abstract, could consider replacing “terrestrial” with this term here in order to really make this point.

We have opted for: “terrestrial (tidal, fluvial, lacustrine and soil)”

Line 49. I think it is sensible to add “their Table 1” after the Davies et al reference. One of the key take home messages in that paper is that MISS are pretty evenly distributed throughout the Ediacaran and Phanerozoic. At present this sentence could mistakenly be misconstrued that Davies et al. hold the school of thought that there is some sort of MISS bias in the Ediacaran. That amended reference on its own would probably be enough here – not sure the relevance of modern salterns in Kolesnikov et al., 2017 here, but realise this was discussed at previous round of review (otherwise why not cite each of the listed MISS occurrences in fluvial-tidal rocks listed in the table in Davies et al.).

We have now followed this recommendation exactly.

Line 60. As commenting on regional place names, a good place to refer to Supp. Figure 1 perhaps?

OK, done.

Line 75. Check convention, but I don’t believe “Group” should be capitalized here.

It is conventional to capitalise any unit name such as Group when referring to a specific and previously mentioned stratigraphic unit. This usage is in line with the International Stratigraphic Guide (even if not followed universally in academia).

Line 79. Is that comma required? Perhaps stylistic preference.

It’s better without — removed.

Line 82. Delete “here”? The sentence already starts with this, and think reads better without the repetition.

**Edited from “structures interpreted here...” to “structures we interpret...”.
Thanks.**

Line 86. Does “Ediacaran” need to be in this heading? No other time periods are being discussed in this manuscript.

Well observed! Removed.

Line 87. “Gibbett”

Corrected, thanks.

Line 90. Check convention, but I don’t believe formation should be capitalized here.

Please see our response to this reviewer’s comment on line 75.

Line 91. Presuming “typical” relates to quantity, perhaps change to “the more abundant” sandstones.

Agreed, done.

Line 108. Not entirely certain this “re-evaluation” statement is fair. Myrow 1995 provides an extensive overview of the regional stratigraphy. I don’t think anywhere is a “shallowing-upwards” model proposed – but to declare the work as “simple” is probably unnecessary considering the work did not attempt a detailed look at the unit’s sedimentology. Better to simply leave this as: “The Gibbett Hill Formation has always been considered permanently subaqueous (REFs), with evidence of emergence not occurring until the Ferryland Head Formation (REFs). The report of this new bedding plane prompts re-evaluation, suggesting at least one cycle of deepening and then shallowing following this deposit.” (or something along these lines). Note Myrow is not in the reference list.

We agree that this was unclear and have revised it along the lines suggested. It now reads:

The Signal Hill Group has previously been described as a “shallow marine to proximal alluvial fan” shallowing-upwards succession (Myrow, 1995). The discovery of this new desiccated horizon refines this model, revealing at least one cycle of deepening and then shallowing after the deposition of this bedding plane.

Myrow has been added to the reference list.

Line 115. Gehling et al. 2009 (?)

We are not sure what is being asked here. The reference given is correct. There is no such paper as Gehling et al. 2009.

Line 117. In my previous review I didn't clock the local importance of recognizing evidence of emergence in the Gibbett Hill Formation, nice.

Thanks. :-)

Line 132. Delete extra space before lamination.

Corrected, thanks.

Line 137 "Soft-sediment"

Corrected to "convolute lamination" as per next comment.

Line 138. Convolute-lamination (with hyphen for consistency with line 132) is a form of soft-sediment deformation. So why specify both? Were other forms of soft-sediment deformation observed? If so specify.

Corrected to just "convolute-lamination"

Line 143. As discussed, suspect tidally-influenced. Certainly envision the relevant desiccated mudstones hosting the streamers and Arumberia as a tidally-influenced alluvial plain. Perhaps "continental" instead of "non-marine"? If simply reporting what the previous researchers have concluded, then I would recommend an additional sentence building on their work based on your observations.

We have reworded the sentence to make it clearer that the "non-marine" interpretation is present in the cited papers. The rest of the existing paragraph adds our own observations and interpretations, and concludes that further work is needed to assess tidal influence, in line with the reviewer's view.

Line 144. Deducing channel planform from preserved aspect ratios is not straightforward, though I would actually suspect that being able to recognize channel margins at a scale subordinate to the "normal" sedimentary outcrop dimensions (and Bread and Cheese Cove at least) is evidence against a braided planform (where autogenic reorganization of the bars, particularly in a sandy system, would lead to the reworking of the preserved channel margins and far more laterally extensive tabular sandstone bodies). This is speculative though with the present evidence. Up to the authors, but my recommendation would be to simply remove this sentence – the bed geometries aren't that laterally extensive in comparison to many channelized/non-channelized outcrops worldwide. Check out the classic work of Gibling (2006) below –

the figures not only emphasize the enormity of other sandstone bodies, but also the extent to which aspect ratios of sandstone bodies might be studied.

• Gibling, M.R., 2006. Width and thickness of fluvial channel bodies and valley fills in the geological record: a literature compilation and classification. *Journal of sedimentary Research*, 76(5), pp.731-770.

We appreciate the suggestion. Obviously there is a degree of relativity in terms like “laterally extensive” but we think we are using it here in a pretty conventional way that does not put undue stress on any particular palaeoenvironmental interpretation, so have elected to retain this wording.

Line 146. Overbank should be one word.

Corrected, thanks.

Line 147. Add “laterally” before transitions.

See below.

Line 149. More distal to the channels? So the authors believe the channelized buff-coloured sandstone units pass down-system into the red-brown Arumberia-bearing siltstones? I don't believe this is the case, with them instead being laterally adjacent environments (channel and alluvial plain respectively). The red-siltstones encase the buff-coloured sandstones demonstrating this. More likely you will pass down system into permanently (or dominantly) subaqueous facies – i.e., the Gibbett Hill Formation.

We have clarified that this facies transitions upwards in the stratigraphy into the finer facies, and that this is an observation. (Naturally this would imply a lateral transition in the original depositional environment by Walther's Law). We have removed reference to “more distal to the channels” as it was confusing and unnecessary for our conclusions. More work is needed to understand fully the relationship between the sand-rich and silt-rich facies but that is clearly outside the scope of this paper.

Line 152. I still find this a quite bizarre choice of reference. Wave ripples are a prominent part of the “floodbasin” facies of Lebeau and Ielpi, but not actually at the sections featured in their study, which are dominantly aeolian. Authors should keep if they think appropriate, but suspect far more relevant to cite some classic sedimentological work actually discussing the formation of wave ripples in some detail: Reineck and Singh *Depositional Sedimentary Environments*, for example.

The Torridonian “floodbasin” deposits described by Lebeau and Ielpi are (we quote): “characterised by tabular beds preserving wave ripples along their boundaries. These beds are for the most part planar-cross stratified, planar-laminated, or wave-ripple laminated, features indicative of splay complexes

prograding into repeatedly flooded ponds.” Thus, this paper provides primary evidence for the relationship between wave ripples and ephemeral braidplain ponds. We remain happy that this is an appropriate citation, and has been used in a way in-keeping with how others in the field have cited this work.

Lines 190 and 201. Why the change from “bedding planes” to “bed tops”? Think the original works better.

We agree with the reviewer and have reverted to “bedding planes” throughout the MS but added “epirelief” and “upwards” where useful to clarify that we are not referring to the undersides (something that a different reviewer requested in the previous round, which is what prompted the use of “bed tops”).

Line 192 – 195. Can this sentence be broken up into two? It’s quite wordy at present considering it is the introduction to the key bedding plane in the study. Took me a couple of passes at it to make sense of it.

We have reworded this sentence to improve understanding

Line 212. Journal preference, but suspect “1” should become “one”

Changed for now but will of course defer to journal editors.

Line 220. Could replace “.” With “:” and then list reasons(?).

We prefer this as it is.

Line 224. There are Manchuriophycus cracks in the Gibbett Hill Formation. The features described here are of course distinct, but if the authors wish to mention Manchuriophycus here and have photographs of these to hand they could consider adding it as a supplementary image.

Interesting observation! We do not have photos of these to hand, however.

Line 231. “Synaeresis”

OK, corrected.

Line 231. Is the second part of this sentence “or indeed dewatering structures or other sedimentary structures” really necessary? Visual dissimilarity alone proves this isn’t the case. Just seems a little odd to specify why not some particular types of sedimentary structures, and then follow this up by saying “or indeed any sedimentary structure”.

We agree that this was oddly worded. We have rephrased it: “...lack the laminae-intersecting infill sediments seen in sedimentary structures such as synaeresis cracks, desiccation cracks, and dewatering structures.”

Line 233. “demonstrate” instead of “show”(?).

Agreed and changed.

Line 235. Perhaps “Weathering may have enhanced the visual contrast between the better indurated,” is more concise.

Agreed and corrected, thanks.

Line 245. “five”(?)

Changed for now but will of course defer to journal editors.

Line 246. Change “above it” to “higher in the section”? Longer but reads better I believe.

Changed to “higher in the stratigraphy” to avoid repeating “section”

Line 254. “the” underlying laminae

Corrected.

Line 281. Perhaps change McMahon et al., 2017 to McMahon et al., 2021. Maybe appropriate also to credit Bland, 1984 here – seeing as this interpretation largely aligns with his earlier viewpoint. • Bland, B.H., 1984. Arumberia Glaessner & Walter, a review of its potential for correlation in the region of the Precambrian–Cambrian boundary. Geological Magazine, 121(6), pp.625-633.

Yes, quite right. We have made these changes.

Line 302. I think the sentence reads better without the “the” before preserved(?).

Agreed and changed.

Line 324. Continental/Terrestrial/Periodically emergent

OK, changed to “terrestrial”.

Line 326. If making a suggestion to someone else’s work, would it not be more appropriate to reword to something like “suggested here to be probable fluvial streamers”? Unless maybe you’ve seen them first hand, in which case could mention as such.

We have reworded as “suggested here to be possible fluvial streamers”.

Line 335. “that” can be observed

OK, “that” added.

Line 344. Really nice.

Thanks!

Line 356. If “tidal” is lost at the earlier description of these, ensure removed here also.

As noted in response to the comment on Line 18 and 80, we now explain why we think they are tidal. But to be appropriately circumspect, we have deleted the word “tidal” here and from similar remarks in the abstract and introduction.

Lines 383 – 384. This sentence needs changing up if it’s being retained.

- Floodplains should be one word.
- Van der Visjell et al., 2019 refers to modern biofilms, as a possible analogue for Precambrian settings in some instances.

Yes, and the way it is cited is consistent with this.

- Ielpi et al., 2018 (as discussed previously) wholly speculate on the possible impact of microbial mats on architecture.
- So as a minimum the sentence should read “We note that microbial communities can increase sediment cohesion on channel-bounding floodplains (van der Visjell et al., 2019) and thus “potentially” also affect the “preserved” architecture of fluvio-tidal systems (Ielpi et al., 2018).

We have adopted this wording entirely, rounded off with the citation recommended below.

- More relevant would be to cite a recent modelling paper led by Muriel Brückner (below) (in addition to or instead of the work by Ielpi et al). Numerical models are employed to study how various stages of plant evolution might affect sedimentary processes and preservation in fluvio-tidal environments. The first “vegetation stage” ran inputs with only microbiota present, with the results suggesting extensive colonization in floodplain tracts at certain points in the fluvial-marine transition zone, with resultant heightened levels of mud deposition facilitating the preservation of more muddy coastal plain facies.
- To my knowledge this is the only study to date that has done something tangible in terms of linking microbial mats to shifts in architecture. o Brückner, M.Z., McMahon, W.J. and Kleinhans, M.G., 2021. Muddying the Waters: Modeling the Effects of Early Land Plants in Paleozoic Estuaries. *Palaios*, 36(5), pp.173-181.
- As I mentioned in my previous review, no studies of the rock record have demonstrably shown an influence of microbial mats on preserved architecture. Mat-influence appears to be below the average norm of deposition and thus any original sedimentological influence is ultimately lost and not preserved.

We have reworded the sentence exactly as proposed, with this new reference included.

Line 402. “studied”

Corrected, thanks.

Line 412. “William” (thanks)

Sorry about that — corrected.

Line 432. “ripples” should be “ripple marks” (preserved not extant)

Corrected, thanks.

Line 455. “Bed top vs Bedding plane”(?)

Fixed.

There are lots of missing references. Myrow 1995, a couple of Gehling papers, Billings, 1872 and more. A complete sweep of the manuscript is required.

Done.

The supplementary note 2, Arumberia nomenclature, is an excellent addition to the manuscript.

Many thanks.

William McMahon

Reviewer 4

McMahon et al. present new fossils from Ediacaran strata in Newfoundland, Canada, that provide evidence of microbial life in emergent/terrestrial settings. The timing of life’s emergence onto the land surface is controversial. Geochemical evidence points to microbial life on land going back far into Precambrian time, but the fossil record is sparse at best. The evidence presented by McMahon et al. then is exciting, particularly since it provides the earliest evidence of an ecological mode of microbial life, “streamers”, that is common today.

The fossil evidence presented is of high quality and I have relatively few comments. A major point of concern for me, the geological setting of the fossils, has already been addressed by the authors in response to other reviewers’ comments. In particular, the latest revision of the manuscript provides a less specific interpretation of the depositional environment, which I think is supported by the available data (although I agree with reviewer 3 that future work should concentrate on further data collection to unravel the precise palaeoenvironmental context of these fossils). The broader interpretation of palaeoenvironment in this version of the manuscript does not diminish

the significance of the finding in my opinion. I think this paper should be published in Proceedings B.

We appreciate these favourable remarks and thank the reviewer for an insightful and attentive response to our paper, which has prompted several corrections and improvements.

I enclose a pdf with minor comments. Diverging from the other reviewers, my two substantive remaining comments refer to (i) the interpretation that these fossils are mineralised in aluminosilicates and the evidence presented to support this, and (ii) the organisation of the Results/Discussion.

Mineralised in aluminosilicates

The authors argue in lines 219–234 that the ribbons have been replaced by clay minerals, but they show little evidence beyond a single thin-section to support this assertion. Are there any geochemical data to support this claim e.g., EDS? Has all the carbon completely gone? Given the context of bacterial-clay interactions citing some of the work of Kurt Konhauser and colleagues seems appropriate.

These useful points are repeated in the line-by-line comments below (re: pages 43 and 45). We respond to them there in detail, outlining our revisions.

Organisation of the Results/Discussion

I think there should be more delineation between the results and discussion sections of the manuscript. The final section of the results conflates reporting/interpretation of the streamers only (lines 167–254) with a discussion of how they might be grouped on a continuum with Arumberia. I think line 256 would mark a convenient point to begin a new section that deals exclusively with the latter point, reducing confusion for the reader. The paper would then report the three types of structure/fossils seen, discuss how two of them may lie on a continuum, before discussing the significance of the trio.

Thanks — we agree with these suggestions. We have inserted a new subheading, “The streamer–arumberimorph spectrum” on (what was) line 256. We have also edited the previous subheading from “Macroscopic biofilm streamers” to “Argillaceous ribbons interpreted as fossil biofilm streamers”, which better reflects the division of what follows into results and then interpretations. This should help to alleviate any confusion.

Further minor comments from Reviewer 4 (extracted from PDF):

Page 34: Why now?

There have been a profusion of recent papers on the identification of Precambrian non-marine/transitional environments, their biota (including

Arumberia), and their chronostratigraphic context. This topic is undoubtedly heating up and there is plenty of work to do. “Ripe for investigation” is admittedly a value claim but we hope it is justified by the contents of the paper itself!

Page 36: I would add ages to this figure and a temporal timescale to orient the reader.

This is in reference to supp. Figure 1. We have added the U-Pb age for the ‘E’ surface to the figure and caption to provide some temporal context. We also added mention to the caption that the Signal Hill Group is wholly Ediacaran (already mentioned in the main text). This constrains the relevant units to between 565 and 540 Ma.

Page 37: What is the precise stratigraphic relationship to the base? At the moment these data float in time.

We have revised the line to “within ~40 m of the base”. There is inbuilt uncertainty in where the base of the Gibbett Hill Fm sits in time because of diachroneity within the depositional system (as mentioned in the body text).

Page 40: What is meant by generally? Can some deviate? By how much? Can this be quantified?

W. McMahon et al. (2021) have measured and commented in some detail on the orientation of Arumberia ridges in respect to palaeocurrent direction elsewhere. Here, though, the palaeocurrent direction can only inexactly be estimated since the ripples are not very well defined. And since the ridges are branching and irregular (shown in Fig 2E) and not simply parallel lines, it cannot be a perfect alignment. Measuring it precisely would not generate useful information for hypothesis testing in the context of this study.

Page 42: Could the authors be more specific? Can this be quantified?

See previous response.

Page 42: In figure 4 it might help orient the reader to label the structures interpreted as the fossil.

Good suggestion. We have added arrows to 4A and 4B for this purpose.

Page 43: Could there be other phosphate sources?

There can be little doubt that the phosphate comes from the remineralization of organic phosphate, and its presence as concretions (not hardgrounds or allochthonous grains) argues for a diagenetic process. At least, that is the proximal source. One could speculate about how much of organic

phosphate was delivered by run-off and how much was generated by local productivity but this would be beyond our scope.

Page 43: Siphonophycus? See Knoll et al. (1991 J. Paleo).

The quality of preservation here is insufficient for formal taxonomy. We cannot even really say that the structures are sheaths/tubes.

Page 43: What is the evidence for this [preservation by clay replacement]? Is there any geochemistry e.g., EDS data to document clay chemistry? Is there no organic matter left at all? **Related:** Page 43 (and 45): Some of the literature by Kurt Konhauser on bacteria-clay interactions may be applicable here e.g., Konhauser et al. (1993 Geology). Also is attachment of pre-existing clays enough? In line 224 the authors argue for authigenic cements. Does this require clay authigenesis? Anderson et al. (2020 Interface Focus) have noted clay templates on Precambrian microfossils including probable cyanobacteria. And Anderson et al. (2011 Geology) have documented them on Ediacaran macrofossils like Chuaria.

These welcome comments along with those of the editor have prompted the following minor revisions to the wording and citations in relation to the mode of preservation:

- 1. Added “three dimensionally” and “authigenic” to the first description of the style of clay replacement on p. 10.**
- 2. Moderated the support for early authigenesis: “Our observations do not necessarily constrain the timing of clay replacement at Ferryland Head” in the main text (page 10); “The timing of clay authigenesis is not clear from our observations” and “Our observations do not exclude [the possibility of late-state aluminosilicification]” in the new Supplementary Note 3.**
- 3. Removed unnecessary instances of “early” before “authigenic” (throughout MS).**
- 4. Incorporated mention of bacteria-clay interactions with appropriate citation to Konhauser’s work: “Organic materials can readily adsorb colloidal clay minerals, trap and bind fine sediment particles, and (partly through bacterial activity) induce clay mineral precipitation, which may ultimately lead to clay replacement (e.g., Konhauser & Urrutia, 1999; Gabbott et al., 2001; Locatelli et al., 2017)” in the new Supplementary Note 3.**
- 5. Added: “No surviving organic matter was observed in our samples during EDX investigations” in the new Supplementary Note 3.**
- 6. Acknowledged that “authigenic clay haloes are reported from a variety of organically preserved Proterozoic and Phanerozoic macro- and microfossils including probable cyanobacteria and eukaryotic algae (e.g., Gabbott et al., 2001, 2017; Anderson et al., 2011; Cai et al., 2012; Anderson**

et al., 2020; Becker-Kerber et al., 2021). In some of these assemblages, however, clay formation appears to have taken place during late-stage volatilisation of organically preserved material with concomitant infilling by aluminosilicates (Becker-Kerber et al., 2021).”

Page 45: Perhaps a new section is required here?

Agreed and done (see “Organisation” comment above).

Page 46: Is this really permineralization?

We used this word in the same way as Brasier & Callow 2009 (ESR) as a synonym for replacement by minerals. But we now notice that this could cause confusion since permineralization is also used to refer to precipitation of minerals in the porosity within organisms (as in wood). To keep thing simple we have replaced the term with “replacement”. Thanks for picking up on this.

Page 46: Anderson et al. (2020 Interface Focus) document clay minerals associated with microfossils representing a variety of phylogenetic clades e.g., green algae, cyanobacteria

This is very relevant, thank you. We have revised the wording to: “The mode of preservation by clay minerals may result from bacterially mediated reactions but is consistent with either bacterial or eukaryotic affinity for the fossils themselves (Konhauser & Urrutia, 1999; Newman et al., 2016; Anderson et al., 2020).”

Page 47: onto

Done